# Rhythmic tapping difficulties in adults who stutter: A deficit in beat perception, motor execution, or sensorimotor integration?

**Anneke Slis, Christophe Savariaux, Pascal Perrier [ID], Maëva Garnier***

CNRS, Grenoble INP, GIPSA-Lab, University Grenoble Alpes, Grenoble, France

* maeva.garnier@gipsa-lab.fr

**Data Availability Statement:** Data cannot be shared publicly because of the confidentiality clause in the ethical approval (CERGA: IRB00010290-2018-10-16-54). Data are available

## Abstract

### Objectives

The study aims to better understand the rhythmic abilities of people who stutter and to identify which processes potentially are impaired in this population: (1) beat perception and reproduction; (2) the execution of movements, in particular their initiation; (3) sensorimotor integration.

### Material and method

Finger tapping behavior of 16 adults who stutter (PWS) was compared with that of 16 matching controls (PNS) in five rhythmic tasks of various complexity: three synchronization tasks — a simple 1:1 isochronous pattern, a complex non-isochronous pattern, and a 4 tap:1 beat isochronous pattern —, a reaction task to an aperiodic and unpredictable pattern, and a reproduction task of an isochronous pattern after passively listening.

### Results

PWS were able to reproduce an isochronous pattern on their own, without external auditory stimuli, with similar accuracy as PNS, but with increased variability. This group difference in variability was observed immediately after passive listening, without prior motor engagement, and was not enhanced or reduced after several seconds of tapping. Although PWS showed increased tapping variability in the reproduction task as well as in synchronization tasks, this timing variability did not correlate significantly with the variability in reaction times or tapping force.

Compared to PNS, PWS exhibited larger negative mean asynchronies, and increased synchronization variability in synchronization tasks. These group differences were not affected by beat hierarchy (i.e., "strong" vs. "weak" beats), pattern complexity (non-isochronous vs. isochronous) or presence versus absence of external auditory stimulus (1:1 vs. 1:4 isochronous pattern). Differences between PWS and PNS were not enhanced or reduced with sensorimotor learning, over the first taps of a synchronization task.

from the GIPSA-lab (contact via Laurent Girin:
laurent.girin@gipsa-lab.fr) for researchers who
meet the criteria for access to confidential data.

**Funding:** This study was funded by the Agence
Nationale de la Recherche (https://anr.fr/fr/)(Project
StopNCo; ANR-14-CE30-0017; PI: MG) The
funders had no role in study design, data collection
and analysis, decision to publish, or preparation of
the manuscript.

**Competing interests:** The authors have declared
that no competing interests exist.

## Conclusion

Our observations support the hypothesis of a deficit in neuronal oscillators coupling in production, but not in perception, of rhythmic patterns, and a larger delay in multi-modal feedback processing for PWS.

## 1 Introduction

Stuttering is a neuro-motor disorder [1,2], characterized by episodes of disfluent speech, containing repeated, extended or blocked sounds, and disrupted rhythmic flow [3]. These perceptual disfluencies have been related to quantitative differences in respiratory, glottal and articulatory behavior of people who stutter (PWS), compared to typical individuals [4–7]. Significant differences in movement duration, movement timing and reaching accuracy have also been reported in upper limb and non-speech orofacial movements [8–11]. Compared to typical speakers, PWS show larger variability and disrupted timing across and within moving components, such as limbs and articulators [12–17], suggesting a timing deficit.

Although the etiology of stuttering is not fully understood yet, evidence suggests that stuttering is related to dysfunctional dopamine receptors and a disrupted basal ganglia-thalamo-cortical network, affecting both motor control and time processing [18,19]. The hypothesis that speech disfluencies in stuttering are caused by a timing deficit [10,20,21] has been explored, behaviorally, by means of finger tapping tasks. In paced tapping tasks, i.e., when tapping in synchrony with an external metronome or musical excerpt [10,20], some of these studies reported a greater tapping variability in PWS, compared to people who do not stutter (PNS) [21,22]. In addition, when tapping along with a metronome marking a simple isochronous sequence, PWS tend to tap more ahead of the beat, i.e., they show a greater "Negative Mean Asynchrony" (NMA) [10,20]. However, another study failed to reveal any difference in variability during sequences of paced tapping tasks [23].

The observed differences in movement behavior potentially originate from deficits at more than one level, since paced tapping involves multiple simultaneous processes, such as the skill to perceive a periodic beat, the capacity to initiate and execute movements to reproduce that beat, and the ability to monitor and update movement timing on-line, using sensory feedback. The sections below define these different processes in more detail, review already available knowledge concerning their possible impairment in PWS and highlight some unresolved issues that still need to be addressed.

### 1.1 Motor delays and variability in the execution of movements

First, evidence suggests that the inaccurate tapping of PWS originates from difficulties at the motor execution stage [24,25], in particular with regard to initiating and sequencing of movements [3], which is also in accordance with a deficient Basal Ganglia [1,18]. Indeed, several studies have observed longer voice reaction times in PWS [26,27] as well as longer reaction times in non-speech tasks involving finger movements [26,28] (see however Reich et al. [29], who did not find significant differences in finger reaction times). In addition, Max et al. [8] reported longer movement durations, peak velocity latencies, and lower peak velocities for finger flexion. Longer durations were also observed between the peak EMG of lip muscles and the speech onset for PWS [13].

To identify difficulties at the motor execution stage, a complicating factor relates to the exact level at which motor execution hampers. Besides the possibility that muscle functioning can be impaired, another potential explanation for the observed movement variability in PWS concerns inaccurate, unstable, or insufficiently activated internal representations [2,25]. Thus, some authors suggested that PWS do not rely on a feedforward and automatized mode of motor control. Instead, they mainly rely on sensory feedback [30–32], inducing additional processing delays, eventually leading to unstable movement behavior of different effectors, especially at fast rate. Supporting this idea, a greater gestural variability was observed in PWS, compared to PNS, not only in the timing of their gestures, but also in their amplitude and target [25,33,34]. Finally, stuttering frequency is also influenced by task complexity [35] and larger differences between PWS and PNS are observed when the task increases in complexity or speed [33,36,37].

One of the hypotheses is that the observed difficulties originate at the level of motor control. In this context, a first objective of the present study is to explore to what extent the increased timing variability and decreased timing accuracy of PWS is related to difficulties in motor planning and execution. In particular, we investigate whether:

- PWS differ from PNS by increased delays and variability when initiating movements and whether these aspects correlate with the degree of tapping accuracy and consistency observed in synchronization tasks or tapping tasks without an external auditory reference.

- PWS show a greater variability, not only in timing, but also in the strength of movements, and whether these two types of variability are correlated.

- the possibly greater variability of PWS is even more enhanced by task complexity, which "pressures" the motor system.

## 1.2 Beat perception and reproduction

"Beat" perception refers to the emergence of an internal representation of periodicity when listening, seeing, or feeling a regular sequence of stimuli [38–42]. One point of view, supported by several theoretical and experimental studies and encompassed under the general term "Oscillators Coupling Hypothesis", suggests that beat perception involves the in phase tuning of endogenous neuronal oscillations in the brain [43–46] (see also [47,48]) in various frequency ranges, with external physical periodic or oscillatory phenomena. Although there is still ongoing debate on this endogenous oscillator entrainment hypothesis [49,50], the observation that steady state-evoked potentials appear in the delta frequency range [0.5–4 Hz] in subjects who were passively listening to a rhythmic sequence at 2.4Hz, provides support for this hypothesis [43,44,51]. In the context of the "Active Sensing" hypothesis applied to auditory perception, Morillon et al. [52,53] have suggested that the tuning of these neuronal oscillators occurring in the delta frequency range in the auditory cortex is modulated by oscillations occurring in the same frequency range in the motor cortex. Thus, the auditory perception of external beats in the delta frequency range [0.5–4 Hz] is expected to be associated with tuned oscillations both in the auditory and motor cortices.

In the framework of coupled oscillators, several authors have suggested that the reduced synchronization accuracy and consistency of PWS during tapping tasks originate from deficient neuronal oscillator coupling affecting time perception and prediction [54,55]. At the behavioral level, coupling deficiencies between neuronal oscillator in the motor cortex are hypothesized to result in the inability to independently produce an isochronous pattern, without the support of external auditory triggers.

A deficit in the coupling mechanism of neuronal oscillations is also theorized to result in the inability to predict and anticipate a repeated periodic stimulus. Etchell et al. [56] showed that, while listening to regular pulses, typical children showed a peak in beta oscillations in the basal ganglia close to stimulus onset–interpreted as an increased attention and prediction of an event at that time–whereas children who stuttered showed a peak after the stimulus occurred. From a behavioral point of view, however, PWS demonstrated Negative Mean Asynchronies in synchronization tasks, like PNS, suggesting that they are able to anticipate external stimuli and that they are not simply "reacting" [10,20].

A subtler deficit of the coupling mechanism is also expected to result in an inaccurate, more variable, and/or drifting reproduction of the period of a previously perceived beat. Studies using synchronization-continuation tasks have reported ambiguous results, however: some showed increased tapping variability in the continuation phase for PWS, compared to PNS [57], whereas others did not observe any significant difference between both groups [21,23].

A final hypothesis is that a deficit in recovering an underlying beat likely results in increased difficulties to add and remove events within a periodic pattern, and therefore to perceive and reproduce complex rhythms, as well as meter, i.e., the hierarchical organization of a rhythmic sequence into "strong" beats and "weaker" ones (a waltz, for instance, is characterized by a triple meter, with a strong initial beat, followed by two weaker ones, whereas a march is duple-metered, with a strong beat every two beats).

In this context, a second objective of the present study is to explore the ability of PWS to perceive and reproduce an intrinsic beat. To identify the level of impairment, the study follows a "differential" and behavioral approach, comparing the performance of PWS and PNS in different rhythmic tasks of varying complexity.

We explore whether PWS differ from PNS in their ability:

- to produce an isochronous pattern independently, without external auditory triggers.

- to predict and anticipate the occurrence of periodic events.

- to reproduce by themselves, without external auditory triggers, an isochronous pattern at a specific tempo, either immediately after passive listening, or after a few seconds of tapping, i.e., after engaging the motor system.

- to perceive and reproduce higher levels of beat organization, like meter, complex non-isochronous patterns, or patterns in which certain pulses are not explicitly marked by external auditory stimuli.

## 1.3 On-line control of movement timing: Dealing with multi-sensory feedback

Perceiving the beat, and then reproducing it, is a first step in tapping along with an external trigger. An additional step involves correctly synchronizing movements to the beat, using sensory feedback for on-line monitoring and correcting timing errors [58–60]. Resulting delays in the pathway linking motor commands and their sensory consequences need to be compensated by the individual who is tapping. A common phenomenon observed in synchronization tapping tasks is the tendency, even in typical individuals, to anticipate the beat, i.e., demonstrating a Negative Mean Asynchrony (NMA) [61]. This phenomenon is influenced by several factors, such as musical experience (NMA shorter in musicians [62]), beat rate (NMA increases when period increases [58,63]), and rhythmic complexity (NMA is reduced in non-isochronous musical excerpts, compared to an isochronous sequence [20]). The NMA also depends on feedback modalities and is reduced when direct auditory feedback is available

compared to information provided by only tactile-kinesthetic feedback [64]. Aschersleben proposed that NMA reflects a slower processing and integration of tactile feedback than auditory or visual feedback [59,65]. In addition to slower processing and integration, this so-called "sensory accumulation" theory further predicts that the magnitude of auditory-tactile delay, and the resulting NMA, depends on stimulation intensity, which, in case of tapping, is hypothesized to concern the tapping force. The NMA is therefore hypothesized to decrease when tactile-kinesthetic feedback in the form of tapping force increases. In line with this, several authors suggested that the greater NMA observed in PWS is related to either a deficit in one sensory modality–in particular, a reduced kinesthetic acuity [30,66,67], or a deficit in multi-sensory integration [20,59].

Based on this knowledge, a third objective of the present study is to further explore the synchronization abilities of PWS, and to better understand whether:

- the larger degree of NMA observed in PWS can be explained by a weaker tapping force, as predicted by the sensory accumulation theory.

- other intra-individual variations in NMA, due to beat strength in particular, correlate with variations in tapping force.

- the difference in tapping variability between PWS and PNS is similar or larger in a synchronization task than the differences observed in a tapping task without external auditory reference, reflecting possible difficulties with sensorimotor integration, additionally involved in synchronization tasks.

### 1.4 Influence of motor engagement and sensorimotor learning

Finally, it is uncertain to what extent the motor system influences or is intrinsically involved in timing processes. Some evidence, however, points toward this possibility. First, some brain activity is observed in motor regions during passive listening to a rhythmic pattern, without any movement [39,68,69], supporting the idea that beat perception intrinsically involves the motor system. Second, the coupling of neuronal oscillations to an external beat frequency, observed in passive listening to rhythm, is enhanced when gestures, like finger tapping, are simultaneously produced [70]. Also, a more accurate and less variable reproduction of an isochronous sequence is observed after tapping along with the pattern, compared to passively listening before tapping [71]. Altogether, these observations support the idea that people build an internal representation of the beat by detecting the periodicity in sensory inputs without actual movement, but that this internal representation is nevertheless consolidated with engaging the motor system.

In that context, the fourth objective of the present study is to explore whether the increased timing variability could be due not so much to a deficit in perceiving and reproducing a beat *per se*, but rather to a deficit in consolidating or updating the sense of the beat with actual motor engagement or sensorimotor learning. These questions are addressed by comparing whether the differences in tapping accuracy and consistency between PWS and PNS, on paced and unpaced tasks, are observed immediately after passive listening to a rhythmic pattern or emerge after several seconds of tapping (In this case, PWS would be expected to improve their accuracy and consistency, whereas PNS would not).

## 2 Material and methods

### 2.1 Participants

16 PWS and 16 PNS were recruited via certified speech language pathologists, word-of-mouth, and social media. The experimental and control group matched in age, gender, and musical

**Table 1. Female and male (F, M) people who stutter (PWS) and not stutter (PNS), with "age" and "musical training" (1: "none", 2: "moderate", 3: "high").** For the PWS, the stuttering severity was both self-evaluated (1: "mild", 2: "moderate", 3: "severe") and evaluated with the SSI-4 Instrument.

| | PWS | | | | | | PNS | | |
|---|---|---|---|---|---|---|---|---|---|
| | Age | Gender | Musical training | SSI-4 score | | Self-evaluated severity | | Age | Gender | Musical training |
| PWS1 | 44 | F | 0 | 17 | (very mild) | 1 | PNS1 | 50 | 0 | 0 |
| PWS2 | 20 | F | 0 | 20 | (mild) | 3 | PNS2 | 20 | 0 | 0 |
| PWS3 | 56 | M | 0 | 16 | (very mild) | 2 | PNS3 | 59 | 0 | 0 |
| PWS4 | 39 | M | 0 | 10 | (very mild) | 2 | PNS4 | 32 | 0 | 0 |
| PWS5 | 54 | M | 0 | 12 | (very mild) | 2 | PNS5 | 51 | 0 | 0 |
| PWS6 | 44 | M | 0 | 19 | (mild) | 1 | PNS6 | 40 | 0 | 0 |
| PWS7 | 42 | M | 0 | 30 | (moderate) | 3 | PNS7 | 39 | 0 | 0 |
| PWS8 | 20 | M | 0 | 19 | (mild) | 3 | PNS8 | 21 | 0 | 0 |
| PWS9 | 48 | M | 0 | 26 | (moderate) | 3 | PNS9 | 46 | 0 | 0 |
| PWS10 | 65 | M | 0 | 26 | (moderate) | 3 | PNS10 | 70 | 0 | 0 |
| PWS11 | 34 | M | 2 | 10 | (very mild) | 2 | PNS11 | 38 | 2 | 2 |
| PWS12 | 27 | F | 1 | 13 | (very mild) | 1 | PNS12 | 25 | 1 | 1 |
| PWS13 | 19 | M | 1 | 19 | (mild) | 2 | PNS13 | 19 | 1 | 1 |
| PWS14 | 35 | M | 1 | 18 | (mild) | 1 | PNS14 | 34 | 1 | 1 |
| PWS15 | 25 | M | 2 | 18 | (mild) | 2 | PNS15 | 24 | 2 | 2 |
| PWS16 | 48 | M | 2 | 34 | (severe) | 3 | PNS16 | 47 | 2 | 2 |
| Average | 35.7 ± 15.3 | | | | | | Average | 36.0 ± 16.4 | | |

training (see Table 1, and section A of the supplementary material, for details on musical training). All speakers were native monolingual speakers of French and did not report any hearing, speaking, voice, or language problems other than developmental stuttering for the experimental group. The project was approved by the local ethics committee of the University Grenoble Alpes (IRB00010290-2018-10-16-54).

## 2.2 Fluency assessment

Participants were asked to self-evaluate their stuttering severity as 'mild,' 'moderate', or 'severe'. Based on a reading task and a picture description task, a speech therapist, specialized in stuttering, also assessed the participants' stuttering severity objectively with the SSI-4 (Stuttering Severity Instrument) [72]. A significant correlation was observed between SSI-4 scores and the self-evaluated severity (see Table 1) (R = 0.57, p = 0.02). Accordingly, the SSI-4 scores were considered for analysis.

## 2.3 Tasks

Five rhythmic conditions were explored: three isochronous, one non-isochronous and one aperiodic rhythm, summarized in Fig 1.

- **1:1_ISO_SYNC—Synchronization task with an isochronous pattern**
  The participants were presented with a simple periodic pattern with an Inter-stimulus Onset Interval (IOI) of 500 ms (i.e., a tempo of 120 BPM). Since a metrical organization of beats (into groups of 2, 3, or 4) arises naturally and automatically when listening to an isochronous sequence of identical tones [73–75], we controlled for that perceptual grouping and induced the perception of quadruple meter, i.e., with a "strong" or accentuated beat sensed every four pulses, the other beats sensed as "weak" or unaccentuated). To achieve this, auditory stimuli

**Fig 1. Summary of the five tasks: 1:1_ISO_SYNC—Synchronization task with a quadruple metered isochronous pattern; 0:1_ISO_REPRO–Reproduction, without any external reference, of a quadruple metered isochronous pattern, after listening passively to it; 1:4_ISO_SYNC: Synchronization task with a quadruple metered isochronous pattern, where only the strong beats (one every four) were marked by an auditory stimulus; NONISO_SYNC—Synchronization task with a quadruple metered non-isochronous pattern; REACT–Reaction task to an unpredictable and aperiodic pattern.** The small lines indicate the metronome beats of an 8- beat cycle. The black dots indicate the auditory stimuli that were played to the participants. The grey triangles indicate the participants finger taps.

were organized into 8-beat cycles, with a metronome click marking the pulse on each beat, and an additional audio beep (Pitch: 1100 Hz; 20 ms) played simultaneously on the first seven beats only (without variations in pitch, loudness, or duration) (see Fig 1). Participants were instructed to listen passively to two cycles of that pattern before they started tapping in synchrony with the beat. For the analysis, the first 8-beat cycle was distinguished from, and compared to the next two 8-beat cycles (2nd and 3rd) to examine a potential effect of sensori-motor learning during the beginning of these tasks.

- ISO_REPRO–Reproduction, without any external reference, of an isochronous pattern, after listening passively to it

  The participants were presented with the same pattern as described for the synchronization task 1:1_ISO_SYNC. After listening passively to two cycles of the pattern, the external auditory metronome stopped, and the participants started tapping as regularly as possible, trying to keep the same pace as in the previously perceived pattern (120 BPM) (see Fig 1). For the analysis, the first cycle of taps was distinguished from, and compared to, the next two cycles (Taps 9 to 24), to explore the performance of internalizing and reproducing the beat after passive listening, and the potential improving effect of motor engagement in reproduction.

- 1:4_ISO_SYNC—Synchronization task with an isochronous pattern, where only the strong beats (one every four) are marked by an auditory stimulus:
  After listening passively to two cycles of the isochronous pattern described earlier in 1:1_ISO_SYNC and ISO_REPRO, the external auditory stimuli were played back every 4 beats only–on the 1st and the 5th beats of the 8- beat cycle, supposed to be perceived as "strong" in a quadruple meter, while the participants started tapping as regularly as possible, trying to keep the same pace as in the previously perceived pattern (see Fig 1). Only the stabilized phase of this task (2nd and 3rd cycles of taps) was considered for analysis.

- NONISO_SYNC—Synchronization task with a quadruple metered non-isochronous pattern:
  The participants were presented with a non-isochronous pattern of seven taps distributed over the 8-beat cycle, still following a quadruple meter and a tempo of 120 BPM. Five of the notes fell "on the beat" (i.e., synchronized with the metronome pulse) while two fell "half the beat" (i.e., exactly in between two metronome pulses) (see Fig 1). Like in 1:1_ISO_SYNC, a metronome click marked the pulse on each beat, while an audio beep played the seven "notes" of the non-isochronous pattern (without variations in pitch, loudness, or duration) (see Fig 1). After listening passively to two cycles of this pattern, participants started to tap in synchrony with the audio beep. In this task again, only the 2nd and 3rd cycles of taps were considered for analysis.

- REACT–Reaction task to an unpredictable and aperiodic pattern:
  The reaction task consisted of responding with a tap as quickly as possible after hearing auditory beeps, played in a non-periodic, and therefore unpredictable, way (see Fig 1). The interstimulus onset interval (IOI) ranged from 200 to 800 ms, with a quasi-flat distribution over a 1 min interval. Unlike in the previous tasks, the REACT task did not include an example phase and the participant could start tapping when ready. In this task, only the 9th to 24th taps were considered for analysis.

The participants performed two trials of each task. The condition 1:1_ISO_SYNC was always performed first, followed by REACT, then NONISO_SYNC. The more complex tasks 1:4_ISO_SYNC and ISO_REPRO were performed at the end of the session. During a practice session outside the booth, the experimenters explained and practiced the tasks with the participants until they were sure that the participants understood the instructions, which did not mean that they were able to achieve the tasks perfectly.

Next, before the actual rhythmic task started, but already inside the booth and being experimentally set-up, spontaneous French speech was elicited by a "spot-the-difference" task during which the participant was instructed to describe differences within pairs of pictures. Finally, a French reading text was employed to elicit more controlled speech material. Both tasks provided material to evaluate the Stuttering Severity Index and to familiarize participants with the experimental setup.

## 2.4 Data collection and experimental set-up

During the experiment, the participants sat at a table, with their dominant lower arm and hand resting on the table, such that they were able to move the index finger easily without moving the arm or hand. Finger tapping events were recorded using a gauge strain sensor (EPL-D11-25P from Meas France), attached to the table, and located just under the index finger of the participant. A microphone simultaneously recorded the resulting audio signal. Both the force signal from the sensor and the audio signal were recorded with a Biopac MP150 acquisition system and the associated *Acknowledge* software, at a sampling rate of 20 kHz, over 16 bits.

The auditory stimuli (metronome click and audio beep) were played binaurally through earphones/earplugs at a comfortable level, indicated by the participant. The earphones and the moderate tapping force prevented the participant from getting direct auditory feedback from their taps. The metronome click and audio beep were also recorded on a second channel of the Biopac system, synchronously with the force signal.

## 2.5 Extracted descriptors

First, the force signal was low-pass filtered (Chebyshev filter, cutoff frequency of 100 Hz, using the function filtfilt in Matlab (R2018b) to extract its envelop, and normalized, based on its maximum value observed in each executed tapping task. For each tap, the first sharp peak of the force signal, corresponding to the tapping instant, was detected automatically (using the Matlab function "findpeaks", with a minimum interpeak distance of 200 ms and a 20% threshold for peak height). These tapping instants were saved in PRAAT [76] annotation files, and were all manually verified and corrected.

From each tapping realization, three measures were extracted, based on the output force signal and the auditory signal played to the participant (see Table 2):

**Table 2. Summary of the seven descriptors considered in this study, depending on the condition.**

| | CONDITION | | | | | | |
| --- | --- | --- | --- | --- | --- | --- | --- |
| | **1:1_ISO_SYNC** | | **1:4_ISO_SYNC** | **NONISO_SYNC** | **ISO_REPRO** | | **REACT** |
| | **1st cycle** | **2nd and 3rd cycles** | **2nd and 3rd cycles** | **2nd and 3rd cycles** | **1st cycle** | **2nd and 3rd cycles** | **2nd and 3rd cycles** |
| **Parameters extracted for each tap** | | | | | | | |
| Reaction Time (RT) | | | | | | | x |
| Phase Angle (PA) | | | | | | | |
| • In general, for all taps | x | x | | | | | |
| • Distinguished for | | | | | | | |
| strong beats | | x | x | x | | | |
| weak beats | | x | x | x | | | |
| taps falling "half beat" | | | | x | | | |
| Tapping Force (TF) | | | | | | | |
| • In general, for all taps | | x | | x | | x | |
| • Distinguished for | | | | | | | |
| strong beats | | x | x | x | | | |
| weak beats | | x | x | x | | | |
| taps falling "half beat" | | | | x | | | |
| **Parameters extracted for each train of taps** | | | | | | | |
| Reaction Time Variability (RT_var) | | | | | | | x |
| Phase Locking Value (PLV) | | | | | | | |
| • In general, for all taps | x | x | | | | | |
| • Distinguished for | | | | | | | |
| strong beats | | x | x | x | | | |
| weak beats | | x | x | x | | | |
| taps falling "half beat" | | | | x | | | |
| Drift in ITI over time | | | | | | x | |
| Coefficient of Variation (CV) | | x | | | x | x | |
| Periodicity Error (PE) | | | | | x | x | |
| Tapping Force Variability (TF_var) | | x | | x | | x | |

- Reaction Time (RT, in ms) was measured in the condition REACT as the time difference (ms) between a tap and the closest preceding auditory stimulus. This value was therefore always positive.

- Phase Angle (PA, in degrees) was measured in the conditions 1:1_ISO_SYNC, 1:4_ISO_-SYNC and NONISO_SYNC, as the angular conversion of Tapping Asynchrony, i.e., the time difference (ms) between a tap and the closest metronome pulse, relatively to the Inter-stimulus onset interval of 500 ms (IOI) (see Eq 1). Tapping asynchrony values were always between -250 ms and +250 ms, so that PA values ranged from -180˚ (completely desynchronized in advance to the auditory stimulus) to +180˚ (completely desynchronized following the auditory stimulus), passing through 0˚ (perfectly synchronized with the auditory stimulus). In the analysis, we distinguished taps that were synchronized with "strong" beats of the 8-beat cycle (the 1$^{st}$ and 5$^{th}$ beats, marked by an auditory stimulus in all three conditions 1:1_ISO_SYNC, 1:4_ISO_SYNC and NONISO_SYNC) from those synchronized with "weak" beats of the cycle (all other beats, marked by an auditory stimulus in 1:1_ISO_SYNC and NONISO_SYNC, but only "internalized" in 1:4_ISO_SYNC), and also from those falling "half-beat" (for the condition NONISO_SYNC only).

$$PA = \text{Asynchrony} * \frac{360}{IOI} \qquad (1)$$

- Tapping Force (TF) was defined as the amplitude of the first sharp peak of the force signal. It was not calibrated in Newtons and was therefore expressed in arbitrary units. However, the same experimental set-up and calibration of the recording equipment was used for all participants, enabling inter-and intra- subject comparisons.

  Six other descriptors were measured over a train of taps:

- Variability in Reaction Time (RT_Var) was measured as the standard deviation of RT values over the taps 9th to 24th in the condition REACT.

- Phase Locking Value (PLV), characterizing the consistency of the stimulus-tap synchrony, was measured over the 2$^{nd}$ and 3$^{rd}$ 8-beat cycles of the conditions 1:1_ISO_SYNC, NONISO_SYNC and 1:4_ISO_SYNC, as well as over the very first 8-beat cycle of the condition 1:1_ISO_SYNC. PLV is defined as the norm of the sum of all the $\vec{PA}$ vectors ($\vec{PA}$ is a unit vector of phase PA in a plane) divided by their number N [22] (see Eq 2). In case the stimulus-tap asynchrony, and therefore the PA values, remain constant over a complete tapping train, the corresponding $\vec{PA}$ vectors align and their sum results in a vector of maximum length (i.e., ideally a PLV of 1). If the stimulus-tap asynchrony, and therefore the PA values, vary considerably from tap to tap, the $\vec{PA}$ vectors points into inconsistent directions and their sum results in a vector of smaller length (i.e., a PLV significantly smaller than 1). In case PA varies a lot, the PLV value can also be very small, due to systematic underestimation or over-estimation of the ITI. PLV values were determined separately for the "strong" and "weak" taps during 1:1_ISO_SYNC, 1:4_ISO_SYNC and NONISO_SYNC, as well as for the taps falling "half-beat" for the condition NONISO_SYNC.

$$PLV = \frac{\|\sum \vec{PA}\|}{N} \qquad (2)$$

- Drift in Inter-Tap Interval over time was evaluated over the first 24 taps during the condition ISO_REPRO. The Inter-Tap Interval (ITI) was defined as the time difference, in ms, between two consecutive taps. A significantly non-null regression slope between the variation of ITI values and the tap number of a train (from 1 to 24) indicated whether the ITI followed a global acceleration (positive slope) or deceleration (negative slope). A non-significant slope indicated that no significant drift occurred over time.

- Coefficient of Variation (CV, in %), was measured as the standard deviation of ITI values, relatively to their mean, over the first cycle of taps (1 to 8) and the next two cycles (taps 9 to 24) of the conditions ISO_REPRO and 1:1_ISO_SYNC.

- Periodicity Error (PE, in ms) was measured again over the first cycle of taps (1–8) and the next two cycles (taps 9 to 24) of the conditions ISO_REPRO, as the time difference, in absolute value, between the "target" period of the previously heard pattern (500 ms) and the actual period of the produced train of taps, i.e., its mean ITI value.

- Variability in Tapping Force (TF_Var): was measured as the standard deviation of tapping force values over the stabilized phase ($2^{nd}$ and 3rd cycles, or taps 9 to 24) of the three conditions 1:1_ISO_SYNC, NONISO_SYNC and ISO_REPRO.

## 2.6 Statistical analysis

The statistical analyses were conducted using the software R [77]. Linear mixed models (using the R package nlme) were used to explore the variation of all the descriptors, except Phase Angle, whose variation was explored with Bayesian circular mixed models (using the R package bpnreg [78]). The variables RT and CV were log-transformed, and the variable PLV was logit transformed.

For linear mixed models, hypotheses about the model's normality and homoscedasticity were validated by looking at the residuals' graphs. When more than one fixed effect was considered in the model, the interaction between them was tested with Likelihood Ratio Tests, and specific contrasts were further examined with Bonferroni adjustments (using the R package "multcomp").

For linear mixed models, the contrast between two conditions was considered significant when $p < .05$. Since Bayesian circular mixed models do not return any p-values, two conditions were considered significantly different if their Highest Posterior Density (HPD) intervals, estimated by the model, did not overlap, or if the HPD interval of their difference did not include zero.

## 3 Results

For the sake of conciseness and clarity, the RESULTS section focuses on the main and most interesting results. Complementary analyses were conducted, in particular to test the correlation between some of the parameters. These non-significant results are available in section B of the Supplementary Material.

## 3.1 Motor delays and variability at the initiation of movements

A first question of this study was whether PWS differed from PNS by increased delays and variability at the initiation of movements, revealing a possible deficiency in movement initiation. To this end, differences between PWS and the PNS in the Reaction Time (RT) and its variability (RT_Var) in the condition REACT were tested, based on the mixed models [log(RT) ~

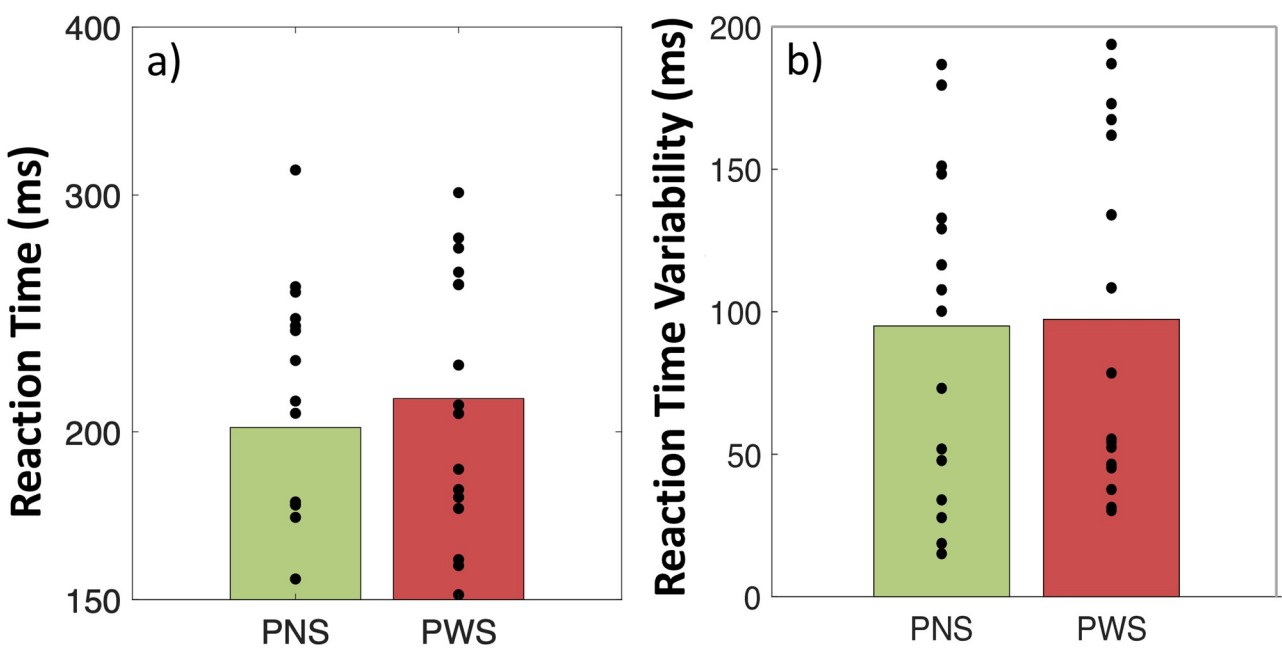

**Fig 2. (a) Average finger reaction time and (b) variability of this reaction time, in the condition REACT during which participants had to follow aperiodic and unpredictible auditory stimuli.** People who stutter (PWS, N = 16) are compared with typical adults without speech disorder (PNS, N = 16).

GROUP + 1|Participant] and [RT_Var ~ GROUP + 1|Participant] (i.e., "GROUP" considered as a fixed effect, and Participant as a random effect).

As expected, a positive Reaction Time, of 232 ± 6 ms on average, was observed when following unpredictable auditory stimuli, as evoked in the condition REACT. PWS and PNS did not differ in their average Reaction Time (F(1,30) = 0.39, p = .54) (see Fig 2a), or in its variability (RT_Var) (F(1, 30) = 0.001, p = .97) (see Fig 2b).

The average RT and RT_Var values of PWS were also not found to correlate significantly with stuttering severity (see section B.1.1 of the supplementary material).

### 3.2 Beat perception and reproduction

**3.2.1 Degree of ITI variability of the reproduced pattern in the condition ISO_RE-PRO.** An additional question was whether PWS faced difficulties with tapping an isochronous sequence on their own, without the help of external auditory triggers. The Coefficient of Variation (CV) of the inter-tap intervals (ITIs) is inversely related to the degree of isochrony of the reproduced pattern in the condition ISO_REPRO. Variations of CV were explored, considering the mixed model [log(CV) ~ GROUP * TIME + 1|Participant], with the 2-level factor TIME = {First 8 beat cycle of ISO_ REPRO; Second and third 8-beat cycles of ISO_REPRO}.

- *Group*: PWS showed a higher Coefficient of Variation (i.e., less isochronous tapping) compared to PNS ($\Delta$log(CV)$_{PNS-PWS}$ = 0.24 ± 0.11, z = 2.19, p = .03) in ISO_REPRO (see Fig 3a). This higher variability of tapping did not correspond to a significant acceleration or deceleration of ITIs is over time (see section B.2.1 of the supplementary Material). The average Coefficient of Variation of PWS was not found to correlate significantly with stuttering severity as well (see section B.2.2 of the supplementary Material).

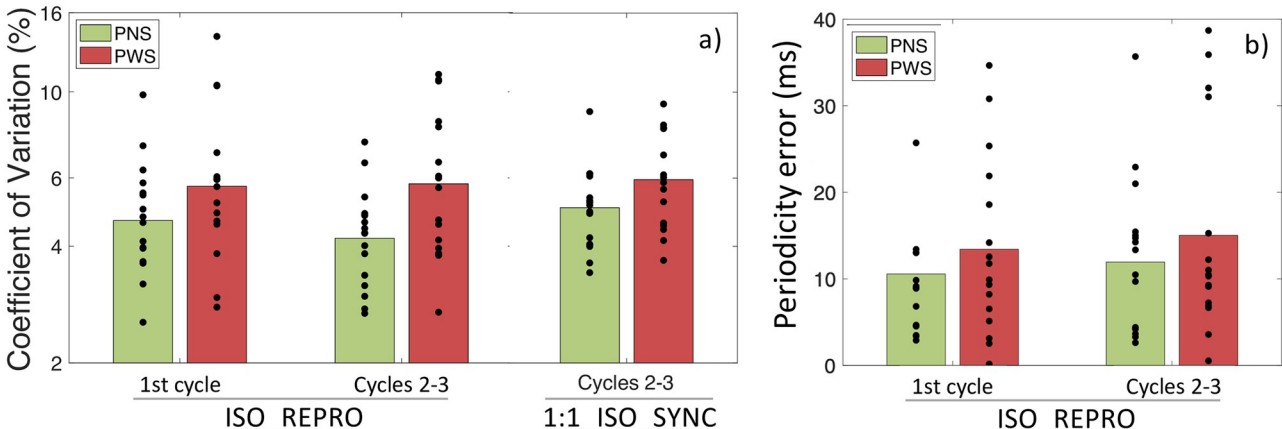

**Fig 3. (a) Coefficient of Variation (CV), inversely related to the degree of isochrony of the reproduced pattern, measured over the very first taps or the stabilized phase of ISO_REPRO, compared to the stabilized phase of the synhcronization task 1:1_ISO_SYNC.** People who stutter (PWS, N = 16) are compared with typical adults without speech disorder (PNS, N = 16). (b) Average Periodicity Error (PE) when reproducing the specific 500ms period of the previously heard isochronous pattern of the condition ISO_REPRO, over the very first taps (first 8-beat cycle) or the more stabilized phase (second and third 8-beat cycles) of the condition.

- *Interaction with motor engagement*: The difference in tapping variability between PNS and PWS, in terms of higher Coefficient of Variation, was observed already during the very first taps (First 8-beat cycle) following passive listening (without motor engagement), and the magnitude of this difference did not change during the second and third cycles, after the motor system had been engaged (No significant interaction GROUP*TIME: df = 1, LRatio:1.28, p = 0.26). For both PWS and PNS, no significant reduction of the Coefficient of Variation was observed between the very first taps and the subsequent ones ($\Delta log(CV)_{9to24-1to8}$ = -0.047 ± 0.051, z = -0.92, p = 0.36) (see Fig 3a).

No significant correlation was observed between the Coefficient of Variation and the variability in Tapping Force (TF_Var) on the same trains of taps, or the variability in Reaction Time (RT_Var) in the REACT condition (see section B.2.2 of the supplementary material).

**3.2.2 Average Periodicity Error (PE) in the condition ISO_REPRO.** One additional question is whether PWS face difficulties at extracting, internalizing, and then reproducing the specific tempo of a periodic pattern. To investigate that question, the variation in Periodicity Error (PE) over a cycle of taps was explored, considering the mixed model [PE ~ GROUP * TIME + 1|Participant], with the 2-level factor TIME = {First 8-beat cycle of ISO_ REPRO; Second and third 8-beat cycles of ISO_REPRO}. No significant interaction GROUP*TIME was observed (df = 1, LRatio = 0.0044, p = 0.95).

- *Group*: PWS were not significantly worse than PNS at reproducing the specific period of the previously heard isochronous pattern ($\Delta PE_{PWS-PNS}$ = 3 ± 3 ms, z = 1.03, p = .30, see Fig 3b): The participants showed an average Periodicity Error of 13 ± 1 ms, which corresponds to 2.6% of the 500 ms IOI target. The average Periodicity Error of PWS was not found to correlate significantly with stuttering severity (see section B.2.3 of the supplementary Material). For both PNS and PWS, this Periodicity Error did not correspond to a systematic under-estimation or over-estimation or the 500 ms pattern period: both groups produced tapping trains with a comparable mean ITI of 501 ± 4 ms ($\Delta Mean\_ITI_{PWS-PNS}$ = 0 ± 5ms, z = 0.07, p = .95). The average Periodicity Error of each participant also did not correlate with his/her

average Reaction Time (RT) in the REACT condition (see section B.2.3 of the supplementary material).

- *Interaction with motor engagement*: Periodicity Error was also not significantly reduced after motor engagement (second and third 8-beat cycles), compared to the first eight-beat cycle of taps following passive listening only ($\Delta PE_{9to24-1to8} = 1 \pm 2$ ms, z = 0.89, p = .37) (see Fig 3b).

### 3.3 Synchronization abilities: Phase Angle (PA: Accuracy) and Phase Locking Value (PLV: Consistency)

**3.3.1 Reference condition (1:1_ISO_SYNC).**   The variations of Phase Angle (PA), Phase Locking Value (PLV) and Tapping Force (TF) in the condition 1:1_ISO_SYNC were explored, considering the Bayesian circular mixed model [PA ~ GROUP + TIME + 1|Participant], and the linear mixed models [logit(PLV) ~ GROUP * TIME + 1|Participant] and [TF ~ GROUP * TIME + 1|Participant], with the 2-level factor TIME = {first 8-beat cycle of 1:1_ISO_SYNC; 2nd and 3rd cycles of 1:1_ISO_SYNC}.

- *Prediction abilities*: Both groups demonstrated negative Phase Angles in 1:1_ISO_SYNC, with an average of -29.4 ± 15.3 degrees (see Fig 4a), which indicated that they were not reacting to the stimulus, as in the condition REACT (see Fig 2a), and that both groups were able to predict and anticipate the beat.

- *Group*: Compared to PNS, PWS showed larger negative Phase Angles, indicating a reduced synchronization accuracy ($\Delta PA_{PWS-PNS} = -10.8 \pm 5.8$ degrees, HPD = [0.2 22.5]) (see Fig 4a), as well as lower Phase Locking Values, signifying a reduced synchronization consistency ($\Delta logit(PLV)_{PWS-PNS} = -0.59 \pm 0.21$, z = -2.81, p < .01) (see Fig 4b). No significant difference in average Tapping Force was observed between PWS and PNS ($\Delta TF_{PWS-PNS} = -0.025 \pm 0.058$ a.u, z = -0.44, p = .66) (see Fig 5). The average Phase Angles and Phase Locking Values of PWS in the condition 1:1_ISO_SYNC were not found to correlate significantly with stuttering severity (see section B.3.1 of the supplementary material).

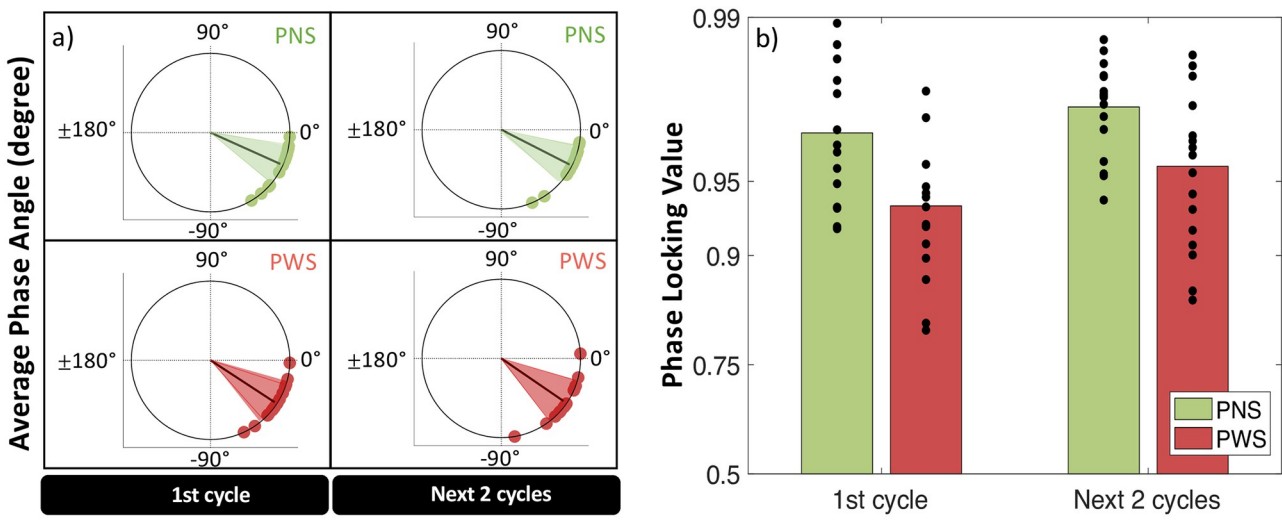

**Fig 4.  (a) Average Phase Angle and (b) Phase Locking Value, for the synchronization task with an isochronous pattern (1:1_ISO_SYNC), over the very first 8-beat cycle of taps or the two next cycles.**

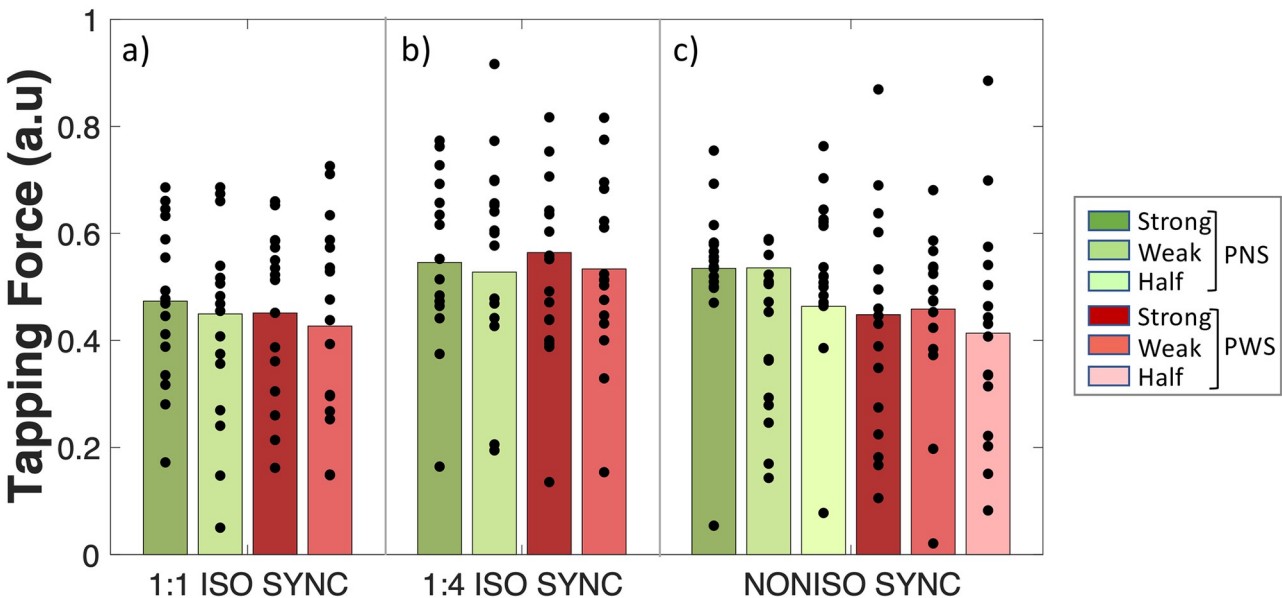

**Fig 5. Tapping Force (in arbitrary unit) on the "strong" vs. "weak" beats of a 8-beat isochronous pattern, in which all the beats were marked by an auditory stiumulus (1:1_ISO_SYNC), or only the strong ones (1:4_ISO_SYNC), and on the "half-beat" pulses of a non-isochronous pattern (NONISO_SYNC).** People who stutter (PWS, N = 16) are compared to with matched control particpants without speech disorders (PNS, N = 16).

- *Interaction with motor engagement*: No significant difference in synchronization accuracy in terms of Phase Angle (PA) was observed between the taps produced during the very beginning of the condition 1:1_ISO_SYNC (first 8-beat cycle), and the next two cycles, for both PNS ($\Delta$PA = 0.9 ± 1.5 degrees, HPD = [-2.1 +4.1]) and PWS ($\Delta$PA = 2.4 ± 2.1 degrees, HPD = [- 1.7 +6.6]) (see Fig 4a). Synchronization consistency, in terms of Phase Locking Values (PLV), however, showed a significant improvement between the first 8-beat cycle and the next two cycles of the condition 1:1_ISO_SYNC ($\Delta$logit(PLV) = 0.33 ± 0.12, z = 2.85, p = .004), for both groups of participants (No significant interaction Time*Group: df = 1, LRatio = 0.35, p = 0.55) (see Fig 4b).

- *Relationship to other indices of motor delays and variability*: The lower Phase Locking Values observed for PWS in this simple synchronization task–revealing an increased variability of the asynchrony between a tap and the closest auditory stimulus–was also related to a greater Coefficient of Variation–corresponding to an increased variability of the inter-tap intervals ($\Delta$log(CV)$_{PWS-PNS}$ = 0.24 ± 0.09, z = 2.53, p = .01) (see Fig 3a). However, this average coefficient of variation in 1:1_ISO_SYNC was significantly greater than in the condition ISO_RE-PRO ($\Delta$log(CV)$_{1:1\_ISO\_SYNC–ISO\_REPRO}$ = 0.13 ± 0.04, z = 3.16, p = .002), for both groups (Non-significant interaction GROUP*TASK: df = 1, LRatio = 3.11, p = 0.08). No significant correlation was observed in 1:1_ISO_SYNC between the degree of NMA and the Tapping Force or between the Phase Locking Values of each train of taps and its corresponding variability in Tapping Force (see section B.3.3 of the supplementary material). No significant correlation was also observed between the average degree of NMA of each participant in 1:1_ISO_SYNC and his/her average reaction time (RT) in the REACT condition, or between the average Phase Locking Value of each participant in 1:1_ISO_SYNC and his/her average Variability in Reaction Time in REACT (see section B.3.2 of the supplementary material).

**3.3.2 Perception and reproduction of meter.** To assess how PWS and PNS perceive and reproduce higher levels of beat organization, the variations of Phase Angle (PA), Phase Locking Value (PLV), and Tapping Force (TF) with metrical hierarchy were further explored, considering for the two tasks, 1:1_ISO_SYNC and NONISO_SYNC, the Bayesian circular mixed models [PA ~ GROUP + STRENGTH + 1|Participant] or the linear mixed models [logit(PLV) ~ GROUP + STRENGTH + 1|Participant] and [TF ~ GROUP + STRENGTH + 1|Participant], with STRENGTH = {strong beats; weak beats} in 1:1_ISO_SYNC, and STRENGTH = {strong beats; weak beats; taps falling "half-beat"} in NONISO_SYNC.

- *Beat strength in 1:1_ISO_SYNC*: The results showed that in the condition 1:1_ISO_SYNC, taps falling on "strong" beats (in our case, the 1st and 5th of each 8-beat cycle) were indeed produced with greater Tapping Force than taps falling on "weak" beats (remaining beats) ($\Delta TF_{strong-weak}$ = 0.019 ± 0.009 a.u, p = .04), for both PWS and PNS (Interaction GROUP*STRENGTH: df = 1, LRatio = 0.003, p = 0.96) (see Fig 5a). They were also synchronized more accurately, i.e., closer to the beat ($\Delta PA_{strong-weak}$ = 4.5 ± 1.8 degrees, HPD = [1.1 8.1]), with a similar strong-weak contrast in both groups of participants (Interaction GROUP*STRENGTH: HPD = [-6.1 +19.8]) (see Fig 6a). Synchronization consistency was not significantly affected by beat strength ($\Delta logit(PLV)_{strong-weak}$ = 0.03 ± 0.19, z = 0.16, p = .87), for both PNS and PWS (Interaction GROUP*STRENGTH: df = 1, LRatio = 0.02, p = .89) (see Fig 7a).

- *Beat strength in NONISO_SYNC*: The significant differences in Phase Angles and Tapping Force observed between strong and weak taps in 1:1_ISO_SYNC, were no longer observed in NONISO_SYNC ($\Delta PA_{strong-weak}$ = -0.7 ± 0.4 degrees, HPD = [-4.7 +3.0]) (see Fig 6c) ($\Delta TF_{strong-weak}$ = 0.010 ± 0.016 a.u, p = .79; see Fig 5c). Like in 1:1_ISO_SYNC, strong and

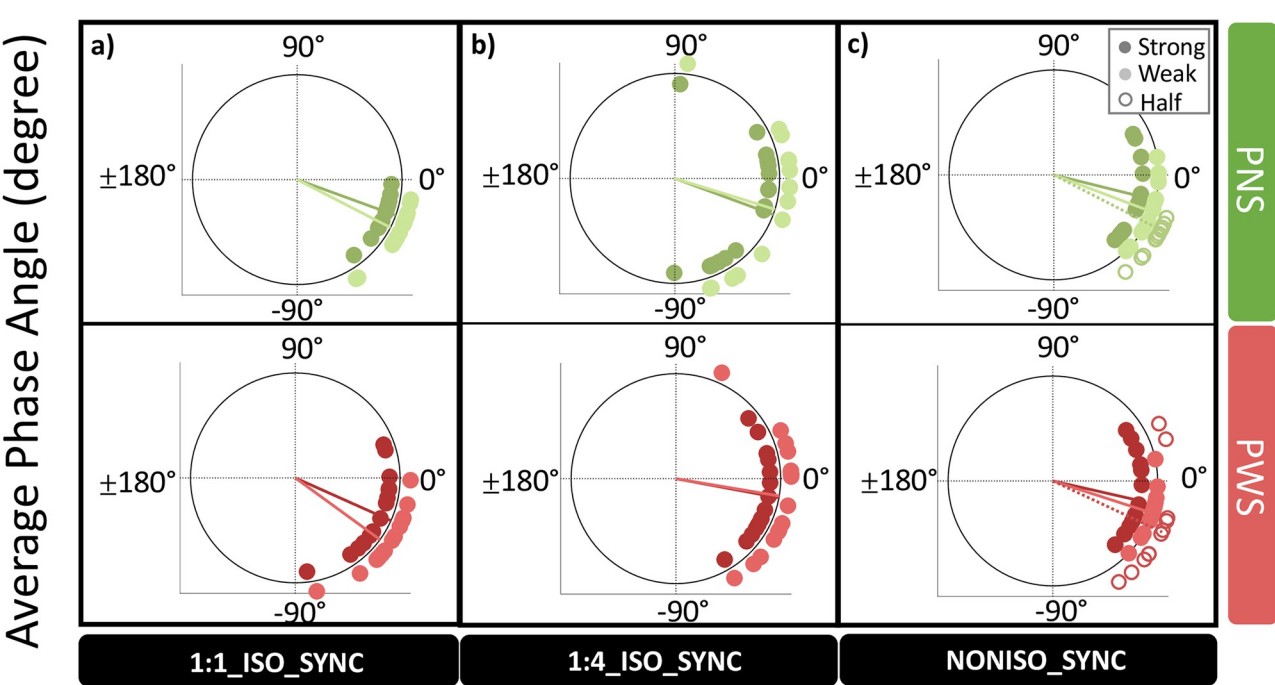

**Fig 6. Average Phase Angle on the "strong" vs. "weak" beats of a 8-beat isochronous pattern, in which all the beats were marked by an auditory stiumulus (1:1_ISO_SYNC), or only the strong ones (1:4_ISO_SYNC), and on the "half-beat" pulses of a non-isochronous pattern (NONISO_SYNC).** People who stutter (PWS, N = 16) are compared to with matched control particpants without speech disorders (PNS, N = 16).

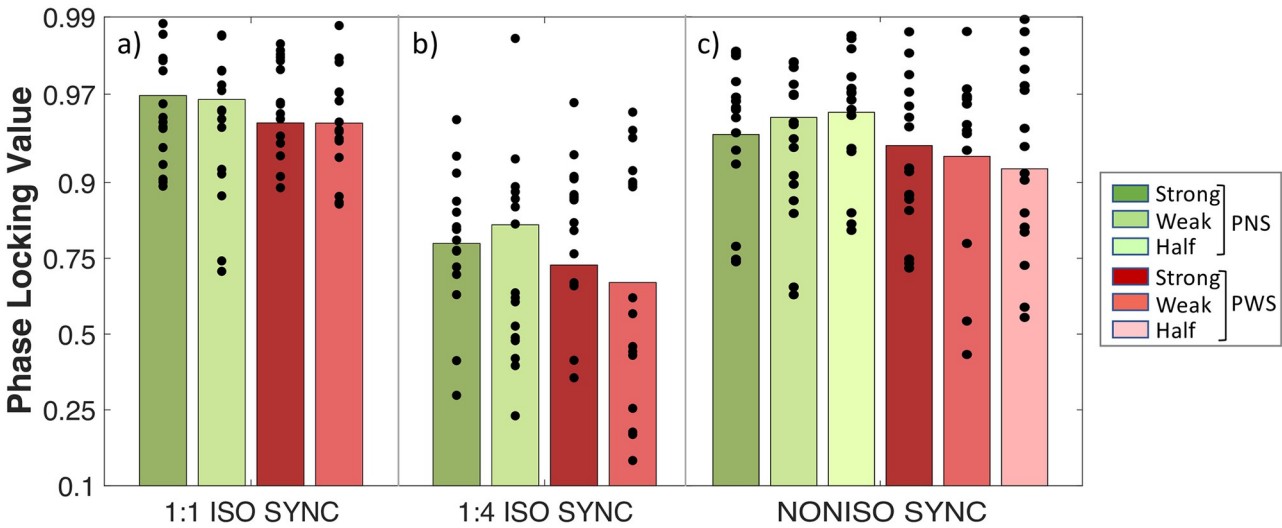

**Fig 7. Phase Locking Value on the "strong" vs. "weak" beats of a 8-beat isochronous pattern, in which all the beats were marked by an auditory stiumulus (1:1_ISO_SYNC), or only the strong ones (1:4_ISO_SYNC), and on the "half-beat" pulses of a non-isochronous pattern (NONISO_SYNC).** People who stutter (PWS, N = 16) are compared to with matched control particpants without speech disorders (PNS, N = 16).

weak taps also did not differ significantly in NONISO_SYNC in terms of synchronization consistency, i.e., Phase Locking Values ($\Delta$logit(PLV)$_{\text{strong-weak}}$ = 0.02 ± 0.13, p = 0.98; see Fig 7c). Taps falling "half-beat" in the condition NONISO_SYNC were also not produced with a significantly reduced consistency, compared to taps falling "on the beat"–synchronized with strong or weak beats ($\Delta$logit(PLV)$_{\text{on the beat-half beat}}$ = 0.01 ± 0.12, z = 0.06, p = .99) (see Fig 7c). However, they were synchronized less accurately ($\Delta$PA$_{\text{on the beat-half beat}}$ = 5.9 ± 2.0, HPD = [2.2 9.9]; Interaction GROUP*STRENGTH: HPD = [-9.2 +21.6])(see Fig 6c), and with a significantly weaker Tapping Force ($\Delta$TF $_{\text{on the beat—half beat}}$ = -0.0730 ± 0.019 a.u, z = -3.87, p = .0002) (see Fig 5c).

**3.3.3 Effect of beat internalization vs. marking by an external auditory stimulus (1:4_ISO_SYNC vs. 1:1_ISO_SYNC).** To investigate how PWS perceive and reproduce internalized beats, the task 1:4_ISO_SYNC in which only the first and fifth beats were marked by an external auditory stimulus was compared to the task 1:1_ISO_SYNC in which a stimulus was played on all beats. The variations of Phase Angle (PA), Phase Locking Value (PLV) and Tapping Force (TF) were further explored in these two tasks, considering the Bayesian circular mixed model [PA ~ GROUP + CONDITION + 1|Participant] and the linear mixed models [logit(PLV) ~ GROUP * CONDITION + 1|Participant] and [TF ~ GROUP * CONDITION + 1|Participant] for either strong or weak beats, distinctly (with CONDITION = {1:1_ISO_-SYNC; 1:4_ISO_SYNC}).

- *Synchronization accuracy*: No significant difference in Phase Angle was observed between 1:1_ISO_SYNC and 1:4_ISO_SYNC for the strong beats, which were marked by an auditory stimulus in both conditions ($\Delta$PA$_{\text{1:4_ISO_SYNC-1:1_ISO_SYNC}}$ = 4.4 ± 6.1 degrees, HPD = [-7.1 +17.0]). This absence of significant differences in accuracy was observed for PWS as well as PNS (No significant Interaction GROUP*CONDITION: HPD = [-16.9 26.1]) (see Fig 6a and 6b). For the condition 1:4_ISO_SYNC, in which weak beats were not marked by an auditory stimulus, both PNS and PWS "synchronized" the weak taps closer to the theoretical beat position, compared to the condition 1:1_ISO_SYNC, in which weak beats were actually

marked by an auditory stimulus ($\Delta PA_{1:4\_ISO\_SYNC-1:1\_ISO\_SYNC}$ = 11.4 ± 3.5 degrees, HPD = [-18.1–4.5]) (No significant Interaction GROUP*CONDITION: HPD = [-23.7 +18.8]).

- *Synchronization consistency*: Phase Locking Values on weak beats were significantly decreased in 1:4_ISO_SYNC, compared to 1:1_ISO_SYNC ($\Delta$logit(PLV)$_{1:4\_ISO\_SYNC-1:1\_ISO\_SYNC}$ = -2.06 ± 0.19, z = -10.89, p < .0001), for PWS as well as PNS (No significant interaction GROUP*CONDITION: df = 1, LRatio = 1.52, p = .22) (see Fig 7a and 7b). A similar decrease in synchronization consistency was observed for strong beats ($\Delta$logit(PLV)$_{1:4\_ISO\_SYNC-1:1\_ISO\_SYNC}$ = -2.10 ± 0.23, z = -0.10, p < .0001), again similarly in PNS and PWS (No significant interaction GROUP*CONDITION: df = 1, LRatio = 0.046, p = .83).

- *Tapping Force*: For both groups, Tapping Force was increased in the condition 1:4_ISO_SYNC, compared to 1:1_ISO_SYNC ($\Delta TF_{1:4\_ISO\_SYNC-1:1\_ISO\_SYNC}$ = 0.086 ± 0.009 a.u, z = 9.83, p < .0001) (see Fig 5a and 5b).

- *Interaction with beat strength*: Finally, the significant difference in synchronization accuracy observed between strong and weak taps in 1:1_ISO_SYNC, was no longer observed in 1:4_ISO_SYNC ($\Delta PA_{strong-weak}$ = 4.9 ± 3.1 degrees, HPD = [-0.7 11.5]) (see Fig 6b). Strong and weak taps also did not differ significantly in synchronization consistency in 1:4_ISO_SYNC ($\Delta$logit(PLV)$_{strong-weak}$ = -0.01 ± 0.18, z = -0.04, p = 0.97), like in 1:1_ISO_SYNC (see Fig 7b). On the contrary, a significant difference in tapping force between strong and weak taps was maintained in 1:4_ISO_SYNC ($\Delta TF_{strong-weak}$ = 0.029 ± 0.012 a.u, z = -2.33, p = .020) for both groups, like in 1:1_ISO_SYNC (see Fig 5b).

**3.3.4 Effect of rhythmic complexity (NONISO_SYNC vs. 1:1_ISO_SYNC).** One of the remaining questions was whether rhythmic complexity enhances the difference in synchronization variability, already observed between PWS and PNS in a simple synchronization task. To this end, the variations of Phase Angle (PA), Phase Locking Value (PLV) and Tapping Force (TF) were also further explored, considering the Bayesian circular mixed model [PA ~ GROUP + CONDITION + 1|Participant] and the linear mixed models [logit(PLV) ~ GROUP * CONDITION + 1|Participant] and [TF ~ GROUP * CONDITION + 1|Participant] for either strong or weak beats, separately (with CONDITION = {1:1_ISO_SYNC; NONISO_SYNC}).

Detailed results are available in section B.3.4 of the supplementary material. In summary, the increased rhythmic complexity in NONISO_SYNC was globally associated with an improved synchronization accuracy (i.e., smaller NMA), compared to the simple synchronization task 1:1_ISO_SYNC (see Fig 6a and 6c), a reduced synchronization consistency (i.e., smaller PLV) (see Fig 7a and 7c), and a greater tapping force (see Fig 5a and 5c). These results were observed for strong as well as weak beats, and similar in PWS and PNS groups.

## 4 Discussion

The study investigated the rhythmic tapping behavior of people who stutter compared to people who do not stutter and considered several levels of processing at which differences were hypothesized to occur: 1- the execution of movements, in particular their initiation (as measured in the task REACT), 2- the perception of beat, at a given periodicity (as measured in the task ISO_REPRO), 3- the on-line adaptation and improvement of their accuracy and consistency, based on sensory feedback (as measured in the tasks 1:1_ISO_SYNC, 1:4_ISO_SYNC and NONISO_SYNC).

## 4.1 Motor delays and variability in the execution of movements

One of the current theories is that stuttering originates from a dysfunctional Basal Ganglia, and more generally a dysfunctional "Cortico-basal ganglia-thalamocortical loop", resulting in disrupted motor execution, such as difficulties initiating movements [3,47]. Several previous studies indeed reported longer voice reaction times [26,27,79] and finger reaction times in PWS [26,28]. Our study did not confirm these studies: no significant difference in average finger reaction time, or its variability, was observed here between PNS and PWS. When taking severity into account, the average reaction time and its intra-individual variance did not correlate significantly with the SSI score. The finger reaction time of each individual did also not predict the average accuracy in a simple synchronization task. No significant link was also observed between the intra-individual variability in reaction time and the consistency at synchronizing with a simple isochronous pattern. These different observations support the idea that, in our experiment, the participants who stutter did not demonstrate a deficit of movement initiation or at least, that this did not affect externally triggered movements.

In their "dual premotor" model, Alm et al. [80] distinguished a "medial" premotor circuit (involving the basal ganglia and the supplementary motor area), involved when initiating and sequencing automatized self-triggered actions, and a "lateral" premotor circuit (involving the cerebellum and the lateral premotor cortex), involved in initiating and sequencing non-automatized actions, triggered by external stimuli. They suggested that the medial circuit is impaired in stutterers, while the lateral one is intact, explaining the observed improved fluency of PWS while speaking with a metronome, choral reading, and singing [81–84]. It was hypothesized that this external information either provides triggers to initiate speech sequences or forces the speaker to pay close attention to the available sensory information, making the movements less automatized. Based on the dual premotor model, we expected in our study to observe timing differences between PWS and PNS, expressed as the Periodicity Error or difference between the dictated inter-stimulus onset interval and the performed interval in the reproduction task ISO_REPRO, during which the tapping was not triggered by external signals, and thus was mediated by the medial premotor circuit only. Contrary to these expectations, no significant difference in Periodicity Error was observed between PNS and PWS. And on the contrary, PWS and PNS did differ significantly in terms of Negative Mean Asynchrony (or Phase Angle) when an external trigger was provided, i.e., in all the synchronization tasks, during which participants were hypothesized to rely more on their lateral premotor circuit. Furthermore, no significant correlation was observed between the average finger reaction time (in REACT) of each individual and his/her average accuracy (PE) in the reproduction task. No significant correlation was also observed between the intra-individual variability in reaction time in REACT and the average tapping variability (CV) in the reproduction task without external auditory stimuli. These results therefore suggest that the timing differences that were further observed between PNS and PWS in our study were not due to motor difficulties regarding initiating movements, whether self-triggered or by an external stimulus.

It was suggested that other possible motor impairments in PWS were associated either with inaccurate internal models, generating instable movements due to larger delays in feedback processing, or with neural noise corrupting the motor commands or the sensory inputs. Both impairments are expected to result in larger variability in motor actions, in terms of movement magnitude, timing or force, for PWS, in addition to greater variability in timing [21,22]. Supporting these hypotheses, previous studies reported a greater variability in movement amplitude and target in PWS [25,33,34], compared to PNS. Although an increased variability in timing was also observed in our study in PWS, when synchronizing with a simple isochronous pattern, no significant difference was observed between the two groups in terms of tapping

force variability for such a "simple" task. In the more complex task NONISO_SYNC, however, an increased variability of the tapping gestures was observed in both timing (decreased PLV) and force, compared to 1:1_ISO_SYNC. Although the group difference in PLV was not significantly enhanced with complexity, PWS showed increased variability of tapping force during complex rhythmic tapping. Finally, no significant correlation was observed between the tapping variability in time (CV) and in force (TF_Var) in the reproduction task, or between the Phase Locking Values and the Tapping Force variability within the simple and complex conditions. These findings suggest that differences in CV and PLV observed between groups (PWS vs. PNS), may not be caused by motor impairment associated with less accurate internal models or by increased neural noise. These results are in line with studies that did not observe a greater motor implementation variance [23] when decomposing the total observed variance in tapping into a motor implementation and a central clock component [85]. This does not mean that PWS may not have any motor difficulties, though, but that these difficulties are not reflected in the tasks that were investigated in our study.

## 4.2 Beat perception and reproduction

In the condition ISO_REPRO, PWS showed the ability to tap an isochronous sequence on their own, without any external auditory reference, i.e., their tapping trains did not show any significant acceleration or slow-down. Furthermore, PWS showed a significant reduction of tapping asynchronies in a predictable pattern (1:1_ISO_SYNC), compared to an unpredictable pattern (REACT), proving their ability to predict and anticipate a regular event. Furthermore, taps were produced with a Periodicity Error (PE) and a tapping variability (CV) that remained within an "acceptable" range, which provides convincing evidence that PWS have the capacity to perceive the specific frequency of a regular pattern, while passively listening to it and to transfer this frequency in the motor domain. In the framework of the Oscillators Coupling Hypothesis, and considering Morillon et al.'s hypotheses [52,53], these observations therefore exclude the hypothesis of a strong deficit in the tuning of neuronal oscillations with the external beat, both in the auditory and in the motor domain, as well as in their interactions [38–42].

Compared to PNS, PWS also did not show a significantly reduced accuracy (i.e., a greater Periodicity Error (PE)) when reproducing a previously perceived isochronous pattern with a specific periodicity. They did, however, show a significantly reduced consistency (i.e., a greater Coefficient of Variation (CV)), which supports the idea that PWS do not have a deficit at perceiving the exact periodicity of an isochronous pattern, but that their difficulties are rather related to reproducing the pattern with tapping gestures.

Several arguments were provided in the preceding section (4.1) that exclude the idea that timing differences between PWS and PNS simply result from an impaired motor execution. However, these differences can possibly be explained in the framework of the Oscillators Coupling Hypothesis, which assumes that neuronal oscillators are tuned in phase and frequency to the frequency of the external periodic stimulation, both in the sensory areas [43,44,51] and in motor areas [52,53]. In our study, the similar level of periodicity error observed in PWS and PNS, but the increased Coefficient of Variation observed in PWS for the condition ISO_RE-PRO (without external auditory triggers), suggests that the perception mechanism of the beat frequency works properly in PWS, in the sense that they *perceive* the beat accurately, but their difficulties are related to a deficit in the coupling of the oscillators driving the motor system,– so that they *reproduce* beat with increased variability. Since, in our study, the higher tapping variability of PWS, compared to PNS, was observed immediately during the first taps of the ISO_REPRO condition, after listening passively to the isochronous pattern to reproduce, and since no significant improvement was observed after several seconds of motor engagement,

the oscillatory coupling deficit in the motor system does not seem to be due to the transition between perception and production, but are instead intrinsic and long lasting.

Finally, since an internalized awareness of beat enables us to link certain rhythmic events as more salient or important than others [86], a deficit in internalizing the beat was hypothesized to result in increased difficulties to perceive and reproduce complex rhythms, as well as to perceive and reproduce meter. In our study, we indeed observed that PWS showed more errors than PNS in the reproduction of the NONISO_SYNC pattern, which is in line with the results of Wieland et al. [87] and supports the idea that PWS have more difficulties than PNS in correctly perceiving and/or reproducing complex non-isochronous patterns. On the other hand, PNS and PWS did not differ significantly in their marking of beat hierarchy: For both PWS and PNS, taps falling on strong beats in 1:1_ISO_SYNC and 1:4_ISO_SYNC, were produced with a greater tapping force, compared with taps falling on weak beats. In NONISO_SYNC, for PNS as well as PWS, taps falling on-beat were also produced with a greater tapping force, compared with taps falling half-beat.

## 4.3 Sensorimotor integration and learning

A significantly reduced timing accuracy and consistency was observed in PWS in synchronization tasks of varying complexity, through greater degrees of Negative Mean Asynchrony and lower Phase Locking Values (PLV). In addition, for both PWS and PNS, the Phase Angles varied with 1- beat strength 2- the presence vs. absence of external auditory stimuli to mark the beat (increased NMA on weak beats in 1:1_ISO_SYNC, in which they are marked by an auditory stimulus, compared to 1:4_ISO_SYNC, in which they are "internalized" by the participants), and 3- task complexity and pulse rate (reduced NMA in both strong and weak beats of NON_ISO_SYNC, compared to 1:1_ISO_SYNC–in agreement with the reduced NMA observed on non-isochronous musical excerpts [20] or with shorter ITI [58,63]). The fact that Phase Angles depended on the task and the lack of correlation with the average Reaction Time (measured in REACT), excludes the idea that NMA in general, and the greater NMA of PWS in particular, correspond to an anticipation strategy aiming at compensating for motor delays at the initiation of movements.

Our results also reject the idea that NMA reflects an under-estimation of Inter-stimulus Onset or Inter-Tap Intervals [20,88]. If this was the case, PWS, who showed a greater NMA, would have demonstrated a global acceleration or an average ITI lower than the pattern's period in the ISO_REPRO condition, when no external stimulus was provided. In this reproduction task, however, PWS did not show any significant drift in ITI over time. The mean ITI of their tapping train was also not systematically "lower" than 500 ms, and comparable to that of PNS.

Many of our observations provide support for a slower processing of tactile and proprioceptive information in PWS, compared to auditory information [59,89]. As a consequence, this reduced kinesthetic sensitivity in PWS [30,66,67] increases this integration delay between auditory and kinesthetic feedback. Several studies indeed show significant differences between PWS and PNS in integrating kinesthetic feedback [30,66,67]. Such a theory explains why PWS perform taps even more in advance to the beat than PNS, so that they more accurately synchronize the perception of the tactile input with the perception of the auditory input. The key-role of synchronizing multiple sensory channels is also compatible with the well-known observation that speech fluency is improved by delayed auditory feedback in PWS [90,91]. If this delayed processing of tactile feedback originates from a systematic slower nerve conduction at the peripheral level ('Fraisse-Paillard' hypothesis [89]), it is predicted that the NMA remains constant within a same individual, regardless of beat strength. This, however, is not the case: in

both groups, we observed that (1) PA is reduced in the weak beats of 1:4_ISO_SYNC, in the absence of acoustic beeps, as compared to the weak beats of 1:1_ISO_SYNC marked with beeps, while this reduction is not observed in strong beats that are marked by acoustic beeps in both tasks; and (2) NMA is reduced in the non-isochronous task (NONISO_SYNC as compared to 1:1_ISO_SYNC). An alternative model, the "sensory accumulation" model [59,65], assumes that the central nervous system detects a sensory stimulation when the number of afferent signals–that increases quickly from the onset of a sensory stimulation, following a so-called "accumulation function"–reaches a certain threshold of sensitivity. The steepness of that accumulation function depends not only on the stimulation intensity, but also on the density of sensory receptors, which is greater for the auditory modality than for the tactile one. Thus, the model assumes that the NMA observed in finger tapping synchronization corresponds to the compensation for the slower accumulation function of tactile feedback received from the finger, compared to that of the auditory metronome stimulation, so that both accumulation functions reach the detection threshold by the central system at the same time. The model furthermore predicts that the amplitude of that auditory-tactile delay, and the resulting NMA, depends on the stimulation intensity. In particular, the NMA is hypothesized to decrease when tapping force increases. In agreement with these predictions, it was observed in our study that in the condition 1:1_ISO_SYNC, strong taps differed from weak taps by both a greater force and a reduced NMA, in the same way that taps falling on the beat differed from half-beat taps in the condition NONISO_SYNC. Furthermore, PNS tapped in the non-isochronous task NONISO_SYNC with both an increased force and a reduced average NMA, compared to the simple synchronization task 1:1_ISO_SYNC.

Following a similar reasoning, the fact that PLV also did not remain constant within an individual but varied with 1- the presence vs. absence of external auditory stimuli to mark the beat (overall decrease in 1:4_ISO_SYNC, compared to 1:1_ISO_SYNC, for all taps) and with 2- complexity and pulse rate (overall decrease in NONISO_SYNC, compared to 1:1_ISO_-SYNC, for all taps), and the lack of correlation with the variability in Reaction Time (RT_Var) or Tapping Force (TF_Var) again excludes the idea that the decreased PLV in PWS simply relates to motor difficulties, either with the initiation of movements, or more generally with their execution. Instead, the greater variability in inter-tap intervals (CV), similarly observed for both PNS or PWS in the synchronization task 1:1_ISO_SYNC, compared to the reproduction task ISO_REPRO, indicates that synchronizing with an auditory stimulus involves an additional source of variability (probably of sensori-motor nature), in addition to just tapping periodically–which already involves timing and motor variabilities. In any case, this increase in tapping variability in a synchronization task was not significantly enhanced in PWS, compared to PNS, so that there does not appear to be a deficit at this stage.

Finally, the variations in synchronization performance over time in 1:1_ISO_SYNC showed that PWS are distinguished from PNS by reduced synchronization accuracy and consistency as soon as the very first taps following a passive listening of the rhythmic pattern, and that synchronization consistency was then significantly improved after a few seconds of synchronous tapping (one 8-beat cycle). However, this improvement was similar for both PNS and PWS, and no such improvement was observed in terms of synchronization accuracy. These observations therefore exclude the hypothesis that timing difficulties in PWS originate from a deficit in sensorimotor learning, to consolidate internal beat representations.

## 5 Conclusions

Following a "differential" and behavioral approach, this study compared the performance of people who stutter (PWS) and people who do not stutter (PNS) in different rhythmic tasks of

various complexity, to better understand the rhythmic deficits of PWS and to identify at which level some cognitive processes might be impaired, leading to the observed differences.

The data were analyzed from three theoretical perspectives: (1) stuttering is associated with motor deficits affecting the initiation and sequencing of movements or the accuracy of movements due to inaccurate predictive models or neural noise; (2) stuttering is associated with impaired coupling between external physical cyclical phenomena and neural oscillators both in perception and movement production, resulting in deficient beat perception and/or reproduction; (3) stuttering is associated with delays in the processing and integration of (multi)sensory feedbacks, resulting in deficient sensorimotor control and synchronization.

The results from our study, exploring a rhythmic deficiency in PWS, point towards (1) a deficit in neural oscillator coupling in production, but not in perception, of rhythmic patterns in PWS, and (2) a larger delay in multi-modal feedback processing for PWS.

## Supporting information

**S1 File. S1 Fig.** (a) Correlation between this average Tapping Asynchrony in the condition REACT (i.e. the average finger Reaction Time) and the SSI score of PWS. (b) Correlation between this reaction time variability and the SSI score of PWS. **S2 Fig.** Tapping Force Variability in the reproduction task of an isochronous pattern, after passive listening (ISO_RE-PRO) and in both tasks of synhcronization to a 4-beat metered isochronous pattern (1:1_ISO_SYNC) or to a non-isochronous pattern (NONISO_SYNC). **S3 Fig.** (a) Correlation between the average log(CV) value and SSI score of PWS. (b) Correlation between the average log(CV) and the Reaction Time Variability (in the condition REACT) of each participant (N = 32). (c) Correlation between log(CV) and the Tapping Force Variability on each train of taps produced in the condition ISO_REPRO. **S4 Fig.** (a) Correlation between the average PE and the SSI score of PWS. (b) Correlation between the average PE in the condition ISO_RE-PRO, and the average Reaction Time in the condition REACT of each participant (N = 32). **S5 Fig.** (a) Correlation between the average PA in the condition 1:1_ISO_SYNC, and the SSI score of PWS. (b) Correlation between the average log(PLV) vaalue in the condition 1:1_ISO_-SYNC, and the SSI score of PWS. **S6 Fig.** (a) Correlation between the average Phase Angle (PA) in the condition 1:1_ISO_SYNC, and the average Reaction Time in the condition REACT of each participant (N = 32). (b) Correlation between the average Phase Locking Value (logit((PLV)) in the condition 1:1_ISO_SYNC, and the average Variability in Reaction Time in the condition REACT of each participant. **S7 Fig.** (a) Correlation between the Phase Angle (PA) and the Tapping Force (TF) of each tap in the conditions 1:1_ISO_SYNC and 1:4_ISO_SYNC. (b) Correlation between the logit(PLV))value and the Tapping Force Variability on each train of taps produced in the conditions 1:1_ISO_SYNC and 1:4_ISO_SYNC. People who stutter (PWS, N = 16) are compared to with matched control participants without speech disorders (PNS, N = 16).
(PDF)

## Acknowledgments

We also thank Silvain Gerber, Ladislas Nalborczyk and Pierre Baraduc for their advice on circular statistics, as well as the two reviewers for their constructive comments.

## Author Contributions

**Conceptualization:** Anneke Slis, Christophe Savariaux, Pascal Perrier, Maëva Garnier.

**Data curation:** Maëva Garnier.

**Formal analysis:** Anneke Slis, Maëva Garnier.

**Funding acquisition:** Maëva Garnier.

**Investigation:** Anneke Slis, Christophe Savariaux, Maëva Garnier.

**Project administration:** Maëva Garnier.

**Supervision:** Maëva Garnier.

**Writing – review & editing:** Anneke Slis, Christophe Savariaux, Pascal Perrier, Maëva Garnier.

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
