## [Decision Letter · Decision Letter 0]

10 Feb 2022

PONE-D-21-37460Rhythmic tapping difficulties in adults who stutter: a deficit in Central Clock and/or Motor Implementation?PLOS ONE

Dear Dr. HUEBER,

Thank you for submitting your manuscript to PLOS ONE. After careful consideration, we feel that it has merit but does not fully meet PLOS ONE’s publication criteria as it currently stands. Therefore, we invite you to submit a revised version of the manuscript that addresses the points raised during the review process. As you will see, both reviewers encountered several major issues with the framing, the methods, the data, and the discussion. I agree with their conclusions that the introduction does not accurately convey the theories used to motivate the study, therefore the rationale is not sound. The methods need significant improvements in clarity and consistent use of terminology, as well as correction of several errors. I also agree that it is statistically unsound to analyze the musicians as groups with insufficient sample sizes, and that reviewer 1's recommendation to remove these groupings entirely is necessary. The discussion also needs to be made more coherent. The reviewers are both experts in the field, and their more minor comments and suggestions also warrant careful attention.

We look forward to receiving your revised manuscript.

Kind regards,

Jessica Adrienne Grahn

Academic Editor

PLOS ONE

Journal Requirements:

“We thank our volunteer subjects and the Agence Nationale de la Recherche for support (Project StopNCo; ANR-14-CE30-0017; PI: Maëva Garnier).”

 “This study was funded by the Agence Nationale de la Recherche (https://anr.fr/fr/)(Project StopNCo; ANR-14-CE30-0017; PI : MG)

5. Please ensure that you refer to Figure 8 in your text as, if accepted, production will need this reference to link the reader to the figure.

Reviewers' comments:

Reviewer's Responses to Questions

**Comments to the Author**

1. Is the manuscript technically sound, and do the data support the conclusions?

Reviewer #1: No

Reviewer #2: Partly

2. Has the statistical analysis been performed appropriately and rigorously? 

Reviewer #1: No

Reviewer #2: No

3. Have the authors made all data underlying the findings in their manuscript fully available?

Reviewer #1: No

Reviewer #2: No

4. Is the manuscript presented in an intelligible fashion and written in standard English?

Reviewer #1: No

Reviewer #2: Yes

5. Review Comments to the Author

Reviewer #1: The authors report a single study that compares timing performance of adults who stutter compared to controls for three tapping tasks that vary in rhythmic complexity. Also considered is the potential role of music training in mediating any differences between groups. A general assumption of the study is that there is larger tapping variability for adults who stutter compared to controls, with the goal of their study to identify / pinpoint the nature of the observed differences, using the Wing & Kristofferson (1973) model as one approach to decomposing tapping variability into separate clock and motor components. The main conclusion is that worse tapping performance of adults who stutter compared to controls is due to both increased central clock variability and increased motor variability.

Strengths

The topic of investigating potential timing deficits in stuttering warrants investigation, as there has been an increasing number of studies that have proposed timing deficits in stuttering, with mixed results. Studies that clarify the nature of timing differences between adults who stutter and controls are certainly needed.

Weaknesses

Although the topic of the study is interesting and warrants investigation, the reported study has a number of significant weaknesses that in my view preclude publication. It is further difficult for me to see how a revision would adequately address these concerns without effectively yielding a new manuscript. Here are the main issues.

First, the methodology for the study is poorly described and motivated. The descriptions of the tasks are very hard to unambiguously interpret and also are non-standard versions of the task(s) to assess timing performance that have been used with the W&K model. With respect to the latter, it is not clear why the authors have chosen to use very non-standard versions of synchronize-continue tapping tasks, which makes it difficult to assess the data in relation to previous work. For example, the standard synchronization-continuation task has individuals synchronize finger taps with an isochronous series of tones, which after a certain number of tones, cut out, and participants continue tapping at the same pace/rate until there is a cue to stop. The version described in the manuscript seems somewhat odd as there are two tones presented (one high and one low), with the high tone indicating that the person should tap and tones are organized in eight element groups. Participants were then asked to tap for a certain number of repetitions of the 8-element pattern, but based on the text this was for only the first seven of the eight low tones (??). The continuation task is then treated as a separate task – and seems as far as I can tell to be separate with participants listening during the synchronization phase and tapping during the continuation phase. In general, this alternative version of the sync-continue task is not well motivated and in general I would recommend using the standard version so that the data are more directly comparable to previous data.

There are further a number of confusing elements of the task and procedure descriptions. For example, in the description of the task order, the synchronization to complex rhythm task is listed twice and the continuation task is not listed at all. This seems like it has to be an error.

Second, I find the description of theory in the introduction to be overly loose/muddled. In this respect, in the consideration of the view that individuals who stutter have a temporal processing deficit, the authors equate the theoretical concept of a central clock and with the generation of an internal beat. This connection is problematic because what is meant by a central clock in the literature is a time interval measuring mechanism that is akin to ‘stop watch’ that measures the duration of each to-be-timed interval independently, with some variance (i.e., in the literature this is viewed as an interval-based timing mechanism) whereas internal-beat generation emphasizes the concept of a rhythm and a beat-based (entrainment) mechanism of timing. It is thus odd to be making connections between (correlated) estimates of central clock variance where each successive interval is estimated independently and synchronization measures where synchronization relies on stimulus-driven entrainment. This feels like comparing ‘apples’ with ‘oranges’.

Third, there are a number of questions about how the dependent variables are measured or the resulting data itself, which reduces confidence in interpreting the results and drawing clear conclusions. As one example, the formula for measuring asynchrony seems like it has an error. As far as I can tell, the authors are taking the asynchrony in msec and then multiplying by 360 and then dividing by 0.5. Dividing by 0.5 is multiplying by 2, so they are taking the raw asynchrony in msec and then multiplying by 720, which would not yield a value between -180 and +180 degrees as claimed. The text also describes “with a 360 modulo”, which does not make any sense to me. Here, I believe what the authors mean is that they took the raw time point of each tap (modulo 500 ms – the inter-beat-interval of the metronome), which if then divided by 500 ms would give a value between 0 and 1; they then would need to rescale between -0.5 and 0.5 and multiply by 360 to get a value between -180 and +180 (or an equivalent procedure).

Note here also that the phase values of -180 and +180 correspond to the same phase, which is relevant for the calculation of mean phase angle. If the authors are simply averaging the phases to calculate mean phase angle, then average -180 and +180 would give a value of zero, which is incorrect. In order to compute mean phase angle, it is necessary to use circular descriptive methods.

With respect to the application of the W&K model to decompose tapping variability into clock and motor components, it is not clear whether the authors linearly de-trended the data before applying the model (removing drift), which is standard for use of this model. The model assumes a stationary time series of produced intervals and the effect of consistent drift is to reduce the negative correlation of successive intervals (reducing the estimate of motor variance – and sometimes yielding negative estimates). Positive correlation of successive intervals is a violation of the model and notably occurred frequently for the PWS group (on 45% of the trials as far as I can tell!). That means that the authors threw out almost half of the PWS data for the W&K part of the study – and completely excluded 4 of the PWS participants (see p. 14).

A further comment on the use of the W&K model is that this model assumes that the mean clock interval is exactly equal to the target inter-tap-interval (in this case 500 ms). The purpose of the synchronization phase is simply to set the clock interval in listeners’ minds. With that said, I find it confusing for the authors to be discussing deviations in the mean produced inter-tap-interval (amount of drift) in terms of the W&K model.

With respect to music training, it is not clear why dance was included as part of the music training measure. Dance is not music training, but an individual with > 5 years of dance and consistent practice would be classified according to the authors’ procedure as having a high level of music training. ??

A final comment about dependent measures is that the authors indicated that they used a peak-picking algorithm in MATLAB to determine peaks, but provide very little additional detail about how this algorithm determines peaks, which can be tricky. Along these lines, did the authors conduct any low pass filtering of the data before picking peaks to remove noise? Details of the peak-picking procedure need to be spelled out so that the reader can better evaluate the method used.

Fourth, I have two general comments about the results that detract and significantly limit the contribution of the work. It is unclear (and problematic) to me why so much of the results and conclusions about the difference in timing performance between PWS and PNS, and the relation between decomposed measures of clock and motor variance and synchronization measures rely on breaking each group down into three musical training categories with very small sample sizes (n = 3) for the moderate and high levels of music training for both the PWS and PNS groups. It is very unlikely that based on such small samples sizes that any differences between levels of music training and any mediating effects are due to self-reported levels of music training, but rather due to a combination of other individual difference factors. For the analyses, at a minimum, it would better to include level of music training as a covariate rather than interpreted as an independent variable, which it is not, and not place so much emphasis on it in the write up and focus on the group comparisons and the main questions of interest. 2. Along these lines, I find it odd that although one on the main conclusions of the study is about the relation/correlation between estimates of central clock and motor variance during continuation tapping with the synchronization measures, none of the graphs show these correlations. Rather, Figures 6, 7, and 8, focus on group comparisons of PWS and PNS for the three levels of music training – which given the very small sample sizes for each level of music training are not very meaningful and unrelated to the main question of interest.

Finally, I do not find the level of precision in the writing up to be up to par for publication. The writing would need to be significantly improved and some sections rewritten to improve precision and clarity to a level that is publication quality.

Reviewer #2: The current study tested tapping abilities of 16 adults who stutter compared to a matched control group across five tapping tasks. The authors aimed to try and tease apart whether individuals who stutter have a deficit in a central clock mechanism or a motor execution deficit. The experiment is very interesting, and it’s great to have these measurements within this relatively large group of stutterers compared to matched controls. However, the presentation and analyses were particularly difficult to follow, and there were some questionable analyses made which make the results and conclusions difficult to interpret. Please see more detailed comments below.

Major Points:

Outline and naming of tasks

It was very difficult to follow the tasks and measures taken, largely because the naming conventions seemed to keep changing, and many different measures were taken and not outlined clearly. Perhaps one way to structure this more clearly would be to have all measures and all measurements in a table for an easy-to-understand overview? Or summarise somehow more clearly in a visual way? The listing of tasks as dot points (e.g., Pg. 8-9) and the listing of all extracted measures (pg. 11 – 14 with various levels depending on task etc) is very difficult to take in. There were numerous naming inconsistencies throughout, for example, line 222, it is unclear what is meant by “metronome and tapping instants”. Is this all tasks? In the discussion (ln 614): greater tapping variability during unpaced tapping is mentioned: is this referring to the synchronization continuation task? Continuation isn’t a pure unpaced tapping measure as they had a cue to begin. I was also wondering why there was no pure unpaced tapping task, as people who stutter have been shown to be aided by an external cue (i.e., as they had in the CONT task at the beginning). Inter-response interval could be more clearly labelled as inter-tap interval to fit with previous literature. These inconsistencies and presentation really need to be improved otherwise it’s very difficult to follow the results.

Musical training

One of the main concerns I had while reading these analyses was with the statistics related to musical training. From Table 1, there are only 3 participants in the “moderate” music training group and 3 participants in the “high” music training group in each stuttering vs. non-stuttering group. All the analyses including musical training are therefore reflecting very few participants, with a big group difference compared to those with “low” music training (n = 10 in each group). These different group sizes are also covered up by bar graphs, and it’s impossible to see the spread of data, and whether there are important outliers. Musical training is also confounded with stuttering severity, as for the “moderate” group, there was 1 very mild stutterer and 2 mild stutterers, and in the “high” group there was 1 very mild, 1 mild, and 1 severe stutterer. I suggest to remove all of these analyses involving musical training. This would also streamline and clarify the results and allow for a focus on the results of interest. Perhaps the authors could instead add some additional, i.e., supplementary material looking at correlations with years of musical training (rather than a categorical, arbitrary grouping measure) and some of the tapping measures, as this would give a more continuous measure. However, I don’t think this should be part of the main analysis or story based on the small sample size. Based on these concerns, many of the conclusions in the discussion are not justified.

Some other small comments about the musical training analyses:

- the measure of musical training is very course, and it is unclear what participants were asked. If musical training was an important aspect of the current study, a more sophisticated measure should have been used, such as the Goldsmiths musical sophistication index. Were there any participants who had more than 5 years of training but were not currently playing? This case does not seem to be captured by the current descriptions.

- Line 378 paragraph: when “musicians” and “non-musicians” are compared – is this group 0 vs. group 1 + 2? Please specify. Then in line 388-389 “highly trained” musicians are mentioned – is this just the one group (with 3 participants?)

Analyses

Were there convergence issues in your linear models? Adding musical training and severity as categorical fixed factors (with three levels each) into your model seems like it would have lots of problems, considering e.g., there are only 3 participants with moderate or high music training, and within each group, for those with high training, 1 is very mild, 1 is mild, and 1 is severe. It doesn’t seem like you have enough data to model these interactions, and I would assume that R will tell you this. The statistical analysis section (2.7) seems to suggest that you could combine all of these factors (musical training + severity) in one model, but I couldn’t find this in the results themselves.

Figures

Individual variation should be displayed in all graphs by including individual data points, and/or better representations of the spread of the data (e.g., box plots, but individual data points would be ideal). This would allow the reader to easily see the spread/variance of data, and also group size differences between bars.

Discussion

The numerous theories presented in the discussion also make for some tough reading, with no strong conclusions being made. It seems in the end that it’s unclear what the results show and how they could be reflected in the different models. Perhaps a clearer summary or more integration across theories is necessary here. The final conclusion that “the dual premotor model and the sensory accumulation model” are compatible with most of the observations didn’t come out easily from the discussion. Some reframing and streamlining seems necessary here.

Paragraph starting line. 658 – starts suggesting that there was support for a global deficit in motor skill. I therefore expected this paragraph to show this. However, the conclusion of the paragraph is that stuttering is NOT caused by differences in motor skill. Please make a topic sentence that is consistent with the evidence presented in the paragraph.

Minor Points:

Abstract: Authors mention that there are three finger-tapping synchronization tasks, but then they list 5. Figure 1 also lists 5. It would be useful throughout to be more consistent with the labelling of each task and order of presentation, to make it easier for the reader to process.

Pg. 3 line 62-63 – lower tapping variability compared to what?

Pg. 4, lines. 66-69 - Can you explain the Wing and Kristofferson method, or rephrase the sentence so the reader isn’t expecting an explanation? Is there a reason it can only be applied on unpaced tapping (ln. 75)?

Pg. 4, lines 83-84: please rephrase, as it currently reads as if the hypothesis itself would significantly contribute to variability than central clock variance.

Pg. 5, lines 91-92 – please fix up this sentence. PWS and what?

Pg. 5, lines 106-107 - Couldn’t central clock variance be related also to motor execution problems?

109 – could tapping force just measure confidence?

Data Cleaning: were any taps excluded from the analysis? E.g., while they were beginning the task? From section 2.6 line 215 it seems that all taps were included? Could this increase variability?

Pg. 15, were CV, CCV, MIV, IRI, Finger RT all in the same model? Aren’t there big collinearities between these measurements? And if there’s only 55% of the PWS group with MIV and CCV calculations, it’s missing a lot of data (The MIV and CCV estimations were considered in the analysis only in these cases, which represented 68% of the tapping trains (82% of the PNS group and 55% for the PWS group) and no single value could be calculated for 4 PWS participants.)

Line 318 – was there a reason not to use the Watson-Williams test here?

Line 325 – why is a generalized linear model suddenly used here? What distribution was used?

Lines 331-335 – then we have coefficients bc and SAM – it’s unclear what this adds to the analysis.

Results

Figure 2: Can you please show individual data in this graph (e.g., as small dots)? Please also mention how many participants are in each group. The phase angles would be better represented as a circular plot in my opinion, e.g., by using the library “circular” in R, or in Matlab using the CircStat toolbox.

Figure 1: from Ln 365-365 it seems that there were both strong and weak beats in SYNCSimp – can you include this information in the figure? Was there an emphasis placed on these beats? Otherwise how are they considered as strong?

The sheer number of acronyms in the results makes it almost impossible to follow at times.

Line 561: Please outline again what REAC means, or use consistent terminology so it’s clear which task is which.

Typos:

Pg. 5 line 91, observed should be observe

Line 95 – “these evidences of” should be “this evidence for”

Line 717: in this line of “though”

Line 760, has two commas.

Please fix up others throughout

6. PLOS authors have the option to publish the peer review history of their article (what does this mean?). If published, this will include your full peer review and any attached files.

Reviewer #1: No

Reviewer #2: No

---

## [Author Response · Author response to Decision Letter 0]

21 Jul 2022

Reviewer #1: 

The authors report a single study that compares timing performance of adults who stutter compared to controls for three tapping tasks that vary in rhythmic complexity. Also considered is the potential role of music training in mediating any differences between groups. A general assumption of the study is that there is larger tapping variability for adults who stutter compared to controls, with the goal of their study to identify / pinpoint the nature of the observed differences, using the Wing & Kristofferson (1973) model as one approach to decomposing tapping variability into separate clock and motor components. The main conclusion is that worse tapping performance of adults who stutter compared to controls is due to both increased central clock variability and increased motor variability.

Strengths

The topic of investigating potential timing deficits in stuttering warrants investigation, as there has been an increasing number of studies that have proposed timing deficits in stuttering, with mixed results. Studies that clarify the nature of timing differences between adults who stutter and controls are certainly needed.

Weaknesses

Although the topic of the study is interesting and warrants investigation, the reported study has a number of significant weaknesses that in my view preclude publication. It is further difficult for me to see how a revision would adequately address these concerns without effectively yielding a new manuscript. Here are the main issues.

• First, the methodology for the study is poorly described and motivated. The descriptions of the tasks are very hard to unambiguously interpret and also are non-standard versions of the task(s) to assess timing performance that have been used with the W&K model. 

The section on task description (L250-299) has been written again to improve its clarity. The introduction now details further the cognitive subprocesses involved in paced and unpaced tapping, in order to better introduce the research questions and to justify the different rhythmic tasks used in this study. Figure 1 gives a visual summary of the different tasks and the new Table 1 recapitulates which parameter was considered in each task.

With respect to the latter, it is not clear why the authors have chosen to use very non-standard versions of synchronize-continue tapping tasks, which makes it difficult to assess the data in relation to previous work. For example, the standard synchronization-continuation task has individuals synchronize finger taps with an isochronous series of tones, which after a certain number of tones, cut out, and participants continue tapping at the same pace/rate until there is a cue to stop.

The continuation task is treated as a separate task – and seems as far as I can tell to be separate with participants listening during the synchronization phase and tapping during the continuation phase. 

In general, this alternative version of the sync-continue task is not well motivated and in general I would recommend using the standard version so that the data are more directly comparable to previous data.

First of all, we modified the label of that task to avoid confusion and comparison with a standard synchronization-continuation task, as used in many previous studies. The condition is now referred to as ISO_REPRO, since it aimed at testing the ability to perceive and reproduce a periodic pattern on one’s own, at a specific tempo. The choice of starting by a passive listening phase, instead of a synchronization phase like in the classical synchronization-continuation paradigm, aimed at distinguishing 1- the ability to build an accurate representation of the beat from passive listening only, without motor engagement, and 2 - the possible “consolidation”, or improvement in accuracy of that representation after several seconds of actual tapping. Our initial hypothesis was that PWS might present a tapping accuracy comparable to that of PNS immediately after passive listening, but that a group difference might emerge only after several seconds of taps, due to a deficient ‘motor consolidation’ mechanism in PWS. The choice of this non-standard task is now better explained in the material & methods section (L 265-274). The underlying theoretical background and the research question, in relationship to neural oscillations, are now more clearly presented in introduction (L108-122+ L194-213). The results section has been re-organized address these questions more explicitly (see L436-492). In particular, additional analyses have been conducted to test the possible improvement in tapping accuracy and consistency after motor engagement, compared to after passive listening only (L451-457 and L489-492).

The version [of the synchronization task] described in the manuscript seems somewhat odd as there are two tones presented (one high and one low), with the high tone indicating that the person should tap and tones are organized in eight element groups. Participants were then asked to tap for a certain number of repetitions of the 8-element pattern, but based on the text this was for only the first seven of the eight low tones (??). 

The label of that task has also been modified to “1:1_ISO_SYNC” and its description has also been improved (L250-264)

The choice of a metered isochronous pattern (instead of a simple metronome) was motivated by 

- Controlling for the metrical organization that would naturally arise when listening to an isochronous sequence (1–3), but that could vary from one listener to another.

- Comparing a simple synchronization task with a more complex one, in which an external stimulus is played every 4 taps only, which clearly induces a perception of quadruple meter. It was therefore necessary to induce the same metrical organization in the first simple task, in which an external stimulus was played on every beat.

- Exploring whether PWS show different abilities with meter compared with PNS, which might indicate a more global and lower level deficit in beat perception.

1. Woodrow H. A quantitative study of rhythm: The effect of variations in intensity, rate and duration. Science Press; 1909. 

2. Fraisse P. Rhythm and tempo. Psychol Music. 1982;1:149–80. 

3. Drake C, Botte MC. Tempo sensitivity in auditory sequences: Evidence for a multiple-look model. Percept Psychophys. 1993;54(3):277–86. 

There are further a number of confusing elements of the task and procedure descriptions. For example, in the description of the task order, the synchronization to complex rhythm task is listed twice and the continuation task is not listed at all. This seems like it has to be an error.

The description of the tasks has been improved and clarified (L250-299), and accompanied by the recapitulative Figure1. The label of the tasks has been changed in order to avoid the confusion between the complexity related to non-isochrony (in the condition now labeled “NONISO_SYNC”) and the complexity related to hearing an external auditory stimulus every 4 taps only (in the condition now labeled “1:4_ISO_SYNC”).

• Second, I find the description of theory in the introduction to be overly loose/muddled. In this respect, in the consideration of the view that individuals who stutter have a temporal processing deficit, the authors equate the theoretical concept of a central clock and with the generation of an internal beat. 

This connection is problematic because what is meant by a central clock in the literature is a time interval measuring mechanism that is akin to ‘stop watch’ that measures the duration of each to-be-timed interval independently, with some variance (i.e., in the literature this is viewed as an interval-based timing mechanism) whereas internal-beat generation emphasizes the concept of a rhythm and a beat-based (entrainment) mechanism of timing. 

It is thus odd to be making connections between (correlated) estimates of central clock variance where each successive interval is estimated independently and synchronization measures where synchronization relies on stimulus-driven entrainment. This feels like comparing ‘apples’ with ‘oranges’.

Thank you for your relevant comment. The introduction has been completely re-written and complemented, following your recommendations. It now details the different subprocesses involved in paced and unpaced tapping, better defines the notions of “beat” and “meter” perception, and the theory of neural oscillators coupling (L108-122). 

The term “central clock” is now avoided, and the correlation between CCV (Central clock variance) and PLV (synchronization consistency) is no longer considered in the new version of the article.

• Third, there are a number of questions about how the dependent variables are measured or the resulting data itself, which reduces confidence in interpreting the results and drawing clear conclusions. As one example, the formula for measuring asynchrony seems like it has an error. As far as I can tell, the authors are taking the asynchrony in msec and then multiplying by 360 and then dividing by 0.5. Dividing by 0.5 is multiplying by 2, so they are taking the raw asynchrony in msec and then multiplying by 720, which would not yield a value between -180 and +180 degrees as claimed. The text also describes “with a 360 modulo”, which does not make any sense to me. Here, I believe what the authors mean is that they took the raw time point of each tap (modulo 500 ms – the inter-beat-interval of the metronome), which if then divided by 500 ms would give a value between 0 and 1; they then would need to rescale between -0.5 and 0.5 and multiply by 360 to get a value between -180 and +180 (or an equivalent procedure). Note here also that the phase values of -180 and +180 correspond to the same phase, which is relevant for the calculation of mean phase angle. If the authors are simply averaging the phases to calculate mean phase angle, then average -180 and +180 would give a value of zero, which is incorrect. In order to compute mean phase angle, it is necessary to use circular descriptive methods.

Our measure of tapping asynchrony, as a phase angle is similar to previous studies (Sares et al. 2019; Falk et al. 2015). The value “0.5” corresponds to the Inter-stimulus Onset Interval (IOI, in seconds). So, dividing the asynchrony (between the stimulus onset and the tapping instant, also in seconds) by the IOI and multiplying it by 360 (degrees) gives a phase angle (in degrees).

We clarified the definition of that descriptor as follows (L 341-355):

“Phase Angle (PA, in degrees) was measured in the conditions 1:1_ISO_SYNC, 1:4_ISO_SYNC and NONISO_SYNC, as the angular conversion of Tapping Asynchrony, i.e. the time difference (ms) between a tap and the closest metronome pulse, relatively to the Inter-stimulus onset interval of 500ms (IOI) (see Eq. 1). By definition, Tapping Asynchrony values were always between -250ms and +250ms, so that PA values ranged from -180° (completely desynchronized in advance to the auditory stimulus) to +180° (completely desynchronized following the auditory stimulus), passing through 0° (perfectly synchronized with the auditory stimulus).“

Bayesian circular mixed models were already used in the previous version of the paper (R package bpnreg). Following your advice, PA results are now also presented with circular plots in the revised version of the manuscript.

• With respect to the application of the W&K model to decompose tapping variability into clock and motor components, it is not clear whether the authors linearly de-trended the data before applying the model (removing drift), which is standard for use of this model. The model assumes a stationary time series of produced intervals and the effect of consistent drift is to reduce the negative correlation of successive intervals (reducing the estimate of motor variance – and sometimes yielding negative estimates). Positive correlation of successive intervals is a violation of the model and notably occurred frequently for the PWS group (on 45% of the trials as far as I can tell!). That means that the authors threw out almost half of the PWS data for the W&K part of the study – and completely excluded 4 of the PWS participants (see p. 14). A further comment on the use of the W&K model is that this model assumes that the mean clock interval is exactly equal to the target inter-tap-interval (in this case 500 ms). The purpose of the synchronization phase is simply to set the clock interval in listeners’ minds. With that said, I find it confusing for the authors to be discussing deviations in the mean produced inter-tap-interval (amount of drift) in terms of the W&K model.

No significant acceleration or deceleration was observed over the 24 first taps of the condition ISO_REPRO in any of the participants (see section B.2.1 of the supplementary material, L40-49).

Nevertheless, we agree that it was problematic that the W&K decomposition was only valid for half of the tapping trains in the PWS group. We therefore decided to remove the analysis based on the W&K model and variance decomposition in the revised version of the article. 

• With respect to music training, it is not clear why dance was included as part of the music training measure. Dance is not music training, but an individual with > 5 years of dance and consistent practice would be classified according to the authors’ procedure as having a high level of music training?

We considered that both the practice of dance and music can influence synchronization abilities, since they consist of producing gestures in synchrony with external rhythms. It has been proven that amongst musicians, drummers and professional pianists show a particularly high synchronization accuracy and consistency, better than singers or non-musicians (Krause, Pollok, and Schnitzler, 2010) (4). Other studies also showed greater synchronization abilities in skilled dancers, compared to non-dancers (Miura, Kudo, Ohtsuki, and Kanehisa, 2011) (5). 

In any case, the 2 dancers in our study also had a corresponding level of musical training. So, to avoid confusion or discussion, we removed any mention to dancing in the assessment of Musical Training (see section A of the supplementary material).

Furthermore, Musical Training was no longer considered as an explicative factor in the revised version of the article. It was only used as an individual factor, like gender and age, to match participants between the PWS and PNS groups.

4. Krause V, Pollok B, Schnitzler A. Perception in action: the impact of sensory information on sensorimotor synchronization in musicians and non-musicians. Acta Psychol (Amst). 2010;133(1):28–37. 

5. Miura A, Kudo K, Ohtsuki T, Kanehisa H. Coordination modes in sensorimotor synchronization of whole-body movement: a study of street dancers and non-dancers. Hum Mov Sci. 2011;30(6):1260–71. 

• A final comment about dependent measures is that the authors indicated that they used a peak-picking algorithm in MATLAB to determine peaks, but provide very little additional detail about how this algorithm determines peaks, which can be tricky. Along these lines, did the authors conduct any low pass filtering of the data before picking peaks to remove noise? Details of the peak-picking procedure need to be spelled out so that the reader can better evaluate the method used.

More details on peak detection are now given on L328-334:

“First, the force signal was low-pass filtered (Chebyshev filter, cutoff frequency of 100 Hz, using the function filtfilt in Matlab (R2018b) to extract its envelop, and normalized, based on its maximum value observed in each executed tapping task. For each tap, the first sharp peak of the force signal, corresponding to the tapping instant, was detected automatically (using the Matlab function “findpeaks”, with a minimum interpeak distance of 200ms and a 20% threshold for peak height). These tapping instants were saved in PRAAT (73) annotation files, and were all manually verified and corrected.”

• Fourth, I have two general comments about the results that detract and significantly limit the contribution of the work. It is unclear (and problematic) to me why so much of the results and conclusions about the difference in timing performance between PWS and PNS, and the relation between decomposed measures of clock and motor variance and synchronization measures rely on breaking each group down into three musical training categories with very small sample sizes (n = 3) for the moderate and high levels of music training for both the PWS and PNS groups. 

It is very unlikely that based on such small samples sizes that any differences between levels of music training and any mediating effects are due to self-reported levels of music training, but rather due to a combination of other individual difference factors. For the analyses, at a minimum, it would better to include level of music training as a covariate rather than interpreted as an independent variable, which it is not, and not place so much emphasis on it in the write up and focus on the group comparisons and the main questions of interest. 

Following your advice, as well as those of the second reviewer, we no longer consider Musical Training as an explicative factor in the revised version of the article. It is now only used as an individual factor, like gender and age, to match participants between the PWS and PNS groups. The figures, the results section and the discussion have also been modified accordingly and do not mention that factor any longer.

• Along these lines, I find it odd that although one of the main conclusions of the study is about the relation/correlation between estimates of central clock and motor variance during continuation tapping with the synchronization measures, none of the graphs show these correlations. Rather, Figures 6, 7, and 8, focus on group comparisons of PWS and PNS for the three levels of music training – which given the very small sample sizes for each level of music training are not very meaningful and unrelated to the main question of interest.

As indicated above, we removed any mention to musical training.

All the correlations tested in the article are now supported by graphs and presented in section B of the supplementary material.

• Finally, I do not find the level of precision in the writing be up to par for publication. The writing would need to be significantly improved and some sections rewritten to improve precision and clarity to a level that is publication quality.

Many sections of the manuscript have been rewritten. 

Furthermore, we had the manuscript proof-read by a native English speaker. 

We therefore hope that it now reaches the expected level of precision and clarity.

 

Reviewer #2: 

The current study tested tapping abilities of 16 adults who stutter compared to a matched control group across five tapping tasks. The authors aimed to try and tease apart whether individuals who stutter have a deficit in a central clock mechanism or a motor execution deficit. The experiment is very interesting, and it’s great to have these measurements within this relatively large group of stutterers compared to matched controls. However, the presentation and analyses were particularly difficult to follow, and there were some questionable analyses made which make the results and conclusions difficult to interpret. Please see more detailed comments below.

Major Points:

• Outline and naming of tasks

It was very difficult to follow the tasks and measures taken, largely because the naming conventions seemed to keep changing, and many different measures were taken and not outlined clearly. Perhaps one way to structure this more clearly would be to have all measures and all measurements in a table for an easy-to-understand overview? Or summarise somehow more clearly in a visual way? The listing of tasks as dot points (e.g., Pg. 8-9) and the listing of all extracted measures (pg. 11 – 14 with various levels depending on task etc) is very difficult to take in. 

The section Material & Methods has been rewritten to improve its clarity. The tasks have been renamed and are summarized in the Figure 1. The extracted measures are now summarized in the Table 1. 

There were numerous naming inconsistencies throughout, for example, line 222, it is unclear what is meant by “metronome and tapping instants”. Is this all tasks?

It is now clarified (L330-333) that tapping instants were detected from the first sharp peak of the force signal, recorded with the force pressure sensor. It is now also clarified (L257-260) that external auditory stimuli, or triggers, were made of both a metronome click marking the pulse, and of audio beeps marking the rhythmic pattern to reproduce. The composition of these external auditory stimuli in each rhythmic condition is summarized in Figure 1.

In the discussion (ln 614): greater tapping variability during unpaced tapping is mentioned: is this referring to the synchronization continuation task? 

The discussion has been fully re-written. Greater tapping variability in PWS has been found in our study as well as most of the previous studies, for both synchronization tasks and tasks in which participants tapped on their own, without any external auditory reference (continuation tasks, reproduction tasks like ISO_REPRO here, or self-paced tasks).

Continuation isn’t a pure unpaced tapping measure as they had a cue to begin. I was also wondering why there was no pure unpaced tapping task, as people who stutter have been shown to be aided by an external cue (i.e., as they had in the CONT task at the beginning).

True. Following your comment, we now avoid the term “unpaced” throughout the article and talk, instead, of tapping without the help of external auditory reference.

Inter-response interval could be more clearly labelled as inter-tap interval to fit with previous literature. These inconsistencies and presentation really need to be improved otherwise it’s very difficult to follow the results.

Done. IRI was changed to ITI.

• Musical training

One of the main concerns I had while reading these analyses was with the statistics related to musical training. From Table 1, there are only 3 participants in the “moderate” music training group and 3 participants in the “high” music training group in each stuttering vs. non-stuttering group. All the analyses including musical training are therefore reflecting very few participants, with a big group difference compared to those with “low” music training (n = 10 in each group). These different group sizes are also covered up by bar graphs, and it’s impossible to see the spread of data, and whether there are important outliers. Musical training is also confounded with stuttering severity, as for the “moderate” group, there was 1 very mild stutterer and 2 mild stutterers, and in the “high” group there was 1 very mild, 1 mild, and 1 severe stutterer. I suggest to remove all of these analyses involving musical training. This would also streamline and clarify the results and allow for a focus on the results of interest. Perhaps the authors could instead add some additional, i.e., supplementary material looking at correlations with years of musical training (rather than a categorical, arbitrary grouping measure) and some of the tapping measures, as this would give a more continuous measure. However, I don’t think this should be part of the main analysis or story based on the small sample size. Based on these concerns, many of the conclusions in the discussion are not justified.

Some other small comments about the musical training analyses: 

- the measure of musical training is very course, and it is unclear what participants were asked. If musical training was an important aspect of the current study, a more sophisticated measure should have been used, such as the Goldsmiths musical sophistication index. Were there any participants who had more than 5 years of training but were not currently playing? This case does not seem to be captured by the current descriptions. 

- Line 378 paragraph: when “musicians” and “non-musicians” are compared – is this group 0 vs. group 1 + 2? Please specify. Then in line 388-389 “highly trained” musicians are mentioned – is this just the one group (with 3 participants?)

Following your advice, as well as those of the first reviewer, we do not consider any longer Musical Training as an independent variable of our experimental design. It is now only used as an individual factor, like gender and age, to match participants between the PWS and PNS groups. The figures, the results section and the discussion have also been modified accordingly and do not mention that factor any longer.

• Analyses

Were there convergence issues in your linear models? Adding musical training and severity as categorical fixed factors (with three levels each) into your model seems like it would have lots of problems, considering e.g., there are only 3 participants with moderate or high music training, and within each group, for those with high training, 1 is very mild, 1 is mild, and 1 is severe. It doesn’t seem like you have enough data to model these interactions, and I would assume that R will tell you this. The statistical analysis section (2.7) seems to suggest that you could combine all of these factors (musical training + severity) in one model, but I couldn’t find this in the results themselves.

The different mixed models considered are now clearly presented at the beginning of each paragraph in the Results section. 

Musical Training is no longer considered as an independent variable. 

Correlations with stuttering severity (though the SSI score) are tested in the PWS group only, and independently from the effect of other factors, such as Group, Task or Beat strength.

• Figures

Individual variation should be displayed in all graphs by including individual data points, and/or better representations of the spread of the data (e.g., box plots, but individual data points would be ideal). This would allow the reader to easily see the spread/variance of data, and also group size differences between bars.

Following your advice, the figures now represent the average value observed for each participant (N=16 in each group), instead of standard deviations with errorbars.

• Discussion

The numerous theories presented in the discussion also make for some tough reading, with no strong conclusions being made. It seems in the end that it’s unclear what the results show and how they could be reflected in the different models. Perhaps a clearer summary or more integration across theories is necessary here. The final conclusion that “the dual premotor model and the sensory accumulation model” are compatible with most of the observations didn’t come out easily from the discussion. Some reframing and streamlining seems necessary here.

The theoretical framework of the whole article has been improved and complemented. Instead of considering two hypotheses: that the reduced tapping accuracy and consistency of PWS may be related to a timing deficit vs. a motor deficit, we now consider different subprocesses that may be involved in sensorimotor synchronization and that may possibly be impaired in PWS (1- motor execution, 2- beat perception and reproduction, 3- sensorimotor integration and learning). The whole article is now organized around the examination of these abilities. This also enables clearer conclusions on the cognitive levels that appear unimpaired, and those at which a significant difference is observed between PWS and PNS. Thus, the results from our study, point towards (1) a deficit in neural oscillators coupling in production, but not in perception, of rhythmic patterns in PWS, and (2) a larger delay in multi-modal feedback processing for PWS.

• Paragraph starting line. 658 – starts suggesting that there was support for a global deficit in motor skill. I therefore expected this paragraph to show this. However, the conclusion of the paragraph is that stuttering is NOT caused by differences in motor skill. Please make a topic sentence that is consistent with the evidence presented in the paragraph.

The discussion section has been fully rewritten. It now makes a clearer distinction between 1- motor difficulties in movement initiation, 2- other possible sources of increased motor variability, and 3- possible deficits in sensorimotor control and learning.

In that framework, our results support the idea that PWS present (1) a deficit in neural oscillators coupling in production, but not in perception, of rhythmic patterns and (2) a larger delay in multi-modal feedback processing 

• Minor Points:

Abstract: Authors mention that there are three finger-tapping synchronization tasks, but then they list 5. Figure 1 also lists 5. It would be useful throughout to be more consistent with the labelling of each task and order of presentation, to make it easier for the reader to process.

There are indeed three synchronization tasks: 

- a simple synchronization task with a 1:1 quadruple metered isochronous pattern (1:1_ISO_SYNC)

- a synchronization task with a quadruple metered isochronous pattern, where only the strong beats (every four taps) are marked by an external auditory stimulus (1:4_ISO_SYNC)

- a synchronization task with a non-isochronous pattern (NONISO_SYNC)

and two other tapping tasks, where participants do not synchronize with auditory stimuli:

- a “reaction” task, in which participants follow as quickly as possible an unpredictable pattern (REACT)

- a reproduction task, in which participants reproduce on their own a quadruple metered isochronous pattern, just after listening to it passively (ISO_REPRO)

Pg. 4, lines. 66-69 - Can you explain the Wing and Kristofferson method, or rephrase the sentence so the reader isn’t expecting an explanation? Is there a reason it can only be applied on unpaced tapping (ln. 75)?

Since the W&K decomposition could not be applied to a substantial part of our data, this analysis was removed from the article and no mention is made to it any longer.

Pg. 5, lines 106-107 - Couldn’t central clock variance be related also to motor execution problems?

The whole idea of the W&K decomposition was to disentangle, from the total tapping variance, what can be attributed to Central Clock Variance (CCV) and what can come from Motor Implementation Variance (MIV). We decided to approach the data differently and remove the analysis from the article, as well as from the discussion of its results.

Pg. 3 line 62-63 – lower tapping variability compared to what?

Pg. 4, lines 83-84: please rephrase, as it currently reads as if the hypothesis itself would significantly contribute to variability than central clock variance.

Pg. 5, lines 91-92 – please fix up this sentence. PWS and what?

The discussion has been fully rewritten.

109 – could tapping force just measure confidence?

Yes, this is possible, and could be used in future studies to evaluate indirectly the difficulty of a task, for example. In this study, it was measured seaking 3 objectives:

1- to test whether participants, and in particular PWS, were able to perceive and reproduce the quadruple meter of the proposed patterns

2- to test the sensory accumulation theory, i.e. whether the degree of NMA correlates with the tapping force

3- to have a descriptor of the amplitude variability of finger movements, and not only of their timing variability

Data Cleaning: were any taps excluded from the analysis? E.g., while they were beginning the task? From section 2.6 line 215 it seems that all taps were included? Could this increase variability?

Following your comment, the revised version of the article now distinguishes the very first taps in a task (first 8-beat cycle) and the “stabilized” phase (2nd and 3rd 8-beat cycles).

Instead of simply excluding the very first taps, we found it relevant, in the two conditions 1:1_ISO_SYNC and ISO_REPRO, to explore the variation in tapping accuracy and consistency over time, between the very first taps and the stabilized phase, and to discuss these variations in terms of motor engagement and sensorimotor learning.

Pg. 15, were CV, CCV, MIV, IRI, Finger RT all in the same model? Aren’t there big collinearities between these measurements? And if there’s only 55% of the PWS group with MIV and CCV calculations, it’s missing a lot of data (The MIV and CCV estimations were considered in the analysis only in these cases, which represented 68% of the tapping trains (82% of the PNS group and 55% for the PWS group) and no single value could be calculated for 4 PWS participants.)

We agree that this was problematic that the W&K decomposition could apply to only half of the tapping trains in the PWS group. We therefore decided to remove any mention of the W&K model and variance decomposition in the revised version of the article.

The article still presents variations in RT, RT_Var = std(RT), mean(ITI), PE = abs(0.5 s – mean(ITI)), and CV = std(ITI)/mean(ITI). 

No significant correlation was observed between CV and RT_Var (see section B.2.2 of the supplementary material).

No significant correlation was also observed between PE and RT (see section B.2.3 of the supplementary material).

The variation of each parameter was explored independently, with different mixed models, as clearly indicated at the beginning of each paragraph in the Results section. 

Line 318 – was there a reason not to use the Watson-Williams test here?

A Watson-Williams test unfortunately did not enable to consider the factor “Participant” underlying the repeated measures of a same individual. Applying a Watson-Williams test over the whole set of “aggregated” data would not be correct, since it would artificially inflate the cohort size (as if we had 32*N repetitions of participants, instead of 32 participants). On the other hand, applying a Watson-Williams test over the 32 average value of each participant would be correct, but would considerably reduce the statistical power of the analysis. Instead, using mixed models (and here Bayesian circular mixed models) is the most recommended method to deal with repeated (angular) data, since it enables to consider the whole dataset, and to account at the same time for the factor “Participant” underlying the intra-individual variation of the dependent variable.

Line 325 – why is a generalized linear model suddenly used here? What distribution was used?

A generalized linear model was used, when testing the correlation between two variables that were measured in different conditions, or when testing the correlation between a variable and the SSI score, since repeated measures could not be considered in that case. Only average values measured for each participant in a condition could be compared, which therefore did not require to use mixed models and to consider a random factor on the participant.

In the revised version of the article, correlations (that are presented in section B of the supplementary material) are simply tested with a Pearson’s correlation test for linear data, and with an angular-linear correlation test (with the toolbox Circstats) for PA.

Lines 331-335 – then we have coefficients bc and SAM – it’s unclear what this adds to the analysis.

In the first version of the manuscript, the reporting of these 3 parameters followed the recommendations of Cremers & Klugkist (2018) for the use of the bpnreg package and the reporting of correlations, based on Bayesian mixed models.

As indicated above, the correlation of PA with other linear variables is now simply tested with an angular-linear correlation test (with the toolbox Circstats) in the revised version of the article.

Results

Figure 2: Can you please show individual data in this graph (e.g., as small dots)? Please also mention how many participants are in each group. 

Done. The size of each group (N=16) is also given in the legend.

The phase angles would be better represented as a circular plot in my opinion, e.g., by using the library “circular” in R, or in Matlab using the CircStat toolbox.

PA variations are now represented with circular plots, following your recommendation.

Figure 1: from Ln 365-365 it seems that there were both strong and weak beats in SYNCSimp – can you include this information in the figure? Was there an emphasis placed on these beats? Otherwise how are they considered as strong?

The description of that task, now labelled 1:1_ISO_SYNC, has been clarified L252-260, and clearly indicates that:

“Since a metrical organization of beats (into groups of 2, 3 or 4) arises naturally and automatically when listening to an isochronous sequence of identical tones (1-3), we controlled for that perceptual grouping and induced the perception of quadruple meter, i.e. with a “strong” or accentuated beat sensed every four pulses, the other beats sensed as “weak” or unaccentuated). To achieve this, auditory stimuli were organized into 8-beat cycles, with a metronome click marking the pulse on each beat, and an additional audio beep (Pitch: 1100 Hz; 20 ms) played simultaneously on the first seven beats only (without variations in pitch, loudness, or duration)

No emphasis was placed on the 1st and 5th beat of the stimuli, which we reckon, is disputable and could be improved in a next study. However, we verified that participants perceived that quadruple metrical organization, since they tapped with greater force on these 1st and 5th beats (see section 3.3.2 and Figure 7). 

The sheer number of acronyms in the results makes it almost impossible to follow at times.

We improved the notation consistency throughout the article, and gave a summary of the task and variable names in the Figure 1 and the Table 1. We hope that it is now easier to follow.

Line 561: Please outline again what REAC means, or use consistent terminology so it’s clear which task is which.

REACT stands for the “reaction” task. The description of the tasks has been improved and summarized in Figure 1.

Typos:

Pg. 5 line 91, observed should be observe

Line 95 – “these evidences of” should be “this evidence for”

Line 717: in this line of “though”

Line 760, has two commas.

Please fix up others throughout

These typos have been fixed or corresponded to paragraphs that have been fully re-written.

---

## [Decision Letter · Decision Letter 1]

12 Sep 2022

PONE-D-21-37460R1Rhythmic tapping difficulties in adults who stutter: a deficit in beat perception, motor execution, or sensorimotor integration?PLOS ONE

Dear Dr. HUEBER,

Thank you for submitting your manuscript to PLOS ONE. After careful consideration, we feel that it has merit but does not fully meet PLOS ONE’s publication criteria as it currently stands. Therefore, we invite you to submit a revised version of the manuscript that addresses the points raised during the review process. You will see that generally the reviewer finds the manuscript easier to follow and has just a few remaining suggestions that I agree might be helpful.

We look forward to receiving your revised manuscript.

Kind regards,

Jessica Adrienne Grahn

Academic Editor

PLOS ONE

Journal Requirements:

Reviewers' comments:

Reviewer's Responses to Questions

**Comments to the Author**

1. If the authors have adequately addressed your comments raised in a previous round of review and you feel that this manuscript is now acceptable for publication, you may indicate that here to bypass the “Comments to the Author” section, enter your conflict of interest statement in the “Confidential to Editor” section, and submit your "Accept" recommendation.

Reviewer #2: (No Response)

2. Is the manuscript technically sound, and do the data support the conclusions?

Reviewer #2: Yes

3. Has the statistical analysis been performed appropriately and rigorously? 

Reviewer #2: Yes

4. Have the authors made all data underlying the findings in their manuscript fully available?

Reviewer #2: No

5. Is the manuscript presented in an intelligible fashion and written in standard English?

Reviewer #2: Yes

6. Review Comments to the Author

Reviewer #2: The current manuscript revision is greatly improved and easier to follow, and the discussion in particular is a lot clearer. It was also nice that the results were condensed to the most interesting, but also that the full analyses were reported in supplementary material. The result structure was easier to follow since it has been condensed, but it was still a bit tricky to track which task and which dependent variable was being analysed. It helped to have the models written out. Perhaps the authors could consider having a more systematic labelling system of headings. At the moment, there are changes in structure, e.g., levels of headings, bolding etc, which make it difficult to clearly see the patterns. Some of the writing is sometimes a bit unclear as well and could be edited further. But overall, the paper is getting into good shape, and the authors have done a good job condensing a lot of dependent variables into a digestible manuscript.

Here are some minor comments to improve clarity:

Pg. 4, lines 64-65: “some other studies” are mentioned, but only one is cited.

Pg. 7: “the observation that steady state-evoked potentials appear in the delta frequency range in subjects who were passively listening to a rhythmic sequence at 2.4Hz provides strong support to this hypothesis”. The existence of steady-state evoked potentials could actually be based on populations of neurons firing in synchrony, not necessarily that they are reflecting the entrainment on endogenous oscillations. There is a big debate about this in the field, so it’s important to clarify this point. E.g., see Zoefel, ten Oever & Sack, 2019: Neural oscillations in the processing of rhythmic input: More than a regular repetition of evoked neural responses. Frontiers in Neuroscience.

Pg. 10, line 185: the greater negative mean asynchrony can be explained by a weaker tapping force – please explain why. The logic behind this is unclear for the moment.

Table 1: musical training is listed as 0, 1, 2 for PWS and as no/yes for PNS – these should be the same scale.

Figure 1: For 1:4_ISO_SYNCH – were the 1st and 4th beats in the example stimuli accented? If so, this should be made clear in the diagram with e.g., an accent marker.

Pg. 16: Only the 9th and 24th taps were considered for analysis. Why? Please motivate the reason for this in the text.

Table 2 is helpful for following the measures taken across the different tasks.

Figure 3 caption seems to be switched around, with 3a as periodicity in the caption, but coefficient of variation in the figure.

Pg. 27 – why was the Bayesian model Group + Time and no interaction?

Pg. 27, line 512: “however, no significant difference…” – I don’t think you need a “however” here. You could just say “there was no significant difference…”.

Pg. 3, line 645 – after explaining all the results across the various measures, it might be nice in this first paragraph to more clearly link the theoretical side with the tasks measured. E.g., 1- in the execution of movements (as measured in xx task/s).

Pg. 36, Lines 739-743: this sentence is very long and refers to already presented information. This could be written more concisely to avoid having to refer to information “already stated above”.

Pg. 40, line 812, do you mean central nervous system?

Pg. 41, line 835: do you mean *decreased* PLV in PWS?

7. PLOS authors have the option to publish the peer review history of their article (what does this mean?). If published, this will include your full peer review and any attached files.

Reviewer #2: No

---

## [Author Response · Author response to Decision Letter 1]

28 Sep 2022

Reviewer #2: The current manuscript revision is greatly improved and easier to follow, and the discussion in particular is a lot clearer. It was also nice that the results were condensed to the most interesting, but also that the full analyses were reported in supplementary material. The result structure was easier to follow since it has been condensed, but it was still a bit tricky to track which task and which dependent variable was being analysed. It helped to have the models written out. 

Perhaps the authors could consider having a more systematic labelling system of headings. At the moment, there are changes in structure, e.g., levels of headings, bolding etc, which make it difficult to clearly see the patterns. 

We worked on this, so that there are now 3 clear levels of headings. A 4 fourth level of organization is sometimes marked by bullet points.

Some of the writing is sometimes a bit unclear as well and could be edited further. 

The entire manuscript has been edited once again by a native speaker of English.

But overall, the paper is getting into good shape, and the authors have done a good job condensing a lot of dependent variables into a digestible manuscript.

Here are some minor comments to improve clarity:

Pg. 4, lines 64-65: “some other studies” are mentioned, but only one is cited.

This was replaced by ‘another study’

Pg. 7: “the observation that steady state-evoked potentials appear in the delta frequency range in subjects who were passively listening to a rhythmic sequence at 2.4Hz provides strong support to this hypothesis”. The existence of steady-state evoked potentials could actually be based on populations of neurons firing in synchrony, not necessarily that they are reflecting the entrainment on endogenous oscillations. There is a big debate about this in the field, so it’s important to clarify this point. E.g., see Zoefel, ten Oever & Sack, 2019: Neural oscillations in the processing of rhythmic input: More than a regular repetition of evoked neural responses. Frontiers in Neuroscience.

We agree with this comment and this paragraph on L111-115 was re-worded as;

“Although there is still ongoing debate on this endogenous oscillator entrainment hypothesis (49,50), the observation that steady state-evoked potentials appear in the delta frequency range [0.5 – 4 Hz] in subjects who were passively listening to a rhythmic sequence at 2.4Hz, provides support for this hypothesis (43,44,51)”, with

[49]. Doelling KB, Assaneo MF, Bevilacqua D, Pesaran B, Poeppel D. An oscillator model better predicts cortical entrainment to music. Proc Natl Acad Sci. 2019;116(20):10113–21. 

[50]. Zoefel B, Ten Oever S, Sack AT. The involvement of endogenous neural oscillations in the processing of rhythmic input: More than a regular repetition of evoked neural responses. Front Neurosci. 2018;12:95. 

Pg. 10, line 185: the greater negative mean asynchrony can be explained by a weaker tapping force – please explain why. The logic behind this is unclear for the moment.

We added an explanation about 10 lines above (L173-177), and clarified the sensory accumulation theory: “In addition to slower processing and integration, this so-called “sensory accumulation” theory further predicts that the magnitude of auditory-tactile delay, and the resulting NMA, depends on stimulation intensity, which, in case of tapping, is hypothesized to concern the tapping force. The NMA is therefore hypothesized to decrease when tactile-kinesthetic feedback in the form of tapping force increases”

We hope that we addressed the confusion re. the logic behind that sentence on L183 “the larger degree of NMA observed in PWS can be explained by a weaker tapping force, as predicted by the sensory accumulation theory”.

Table 1: musical training is listed as 0, 1, 2 for PWS and as no/yes for PNS – these should be the same scale.

The PNS and PWS were matched in musical training. The notation is modified in the table, accordingly.

Figure 1: For 1:4_ISO_SYNCH – were the 1st and 4th beats in the example stimuli accented? If so, this should be made clear in the diagram with e.g., an accent marker.

In that condition, the first and fifth beats were not “accentuated” but were the only ones marked with an external stimulus.

We explained this on P15, 271-272, in the description of the task 1:4_ISO_SYNC, as follows: “the external auditory stimuli were played back every 4 beats only – on the 1st and the 5th beats of the 8- beat cycle”.

In addition, this is also illustrated in the recapitulative Figure 1.

Pg. 16: Only the 9th and 24th taps were considered for analysis. Why? Please motivate the reason for this in the text.

We did not consider the 9th and 24th taps, but the taps number 9 to 24, i.e., the 2nd and 3rd 8-beat cycles of taps, taken as the “stabilized” phase of the task. It was hypothesized that some sensorimotor adaptation and learning might occur during the first cycle of 8 taps and carried over to the remainder of the trial.

The word “Taps 9-24” was replaced by “Taps 9 to 24” to avoid confusion

Table 2 is helpful for following the measures taken across the different tasks.

We are happy that it improved the paper.

Figure 3 caption seems to be switched around, with 3a as periodicity in the caption, but coefficient of variation in the figure.

Modified

Pg. 27 – why was the Bayesian model Group + Time and no interaction?

The notation in the bpnreg package (for circular mixed models) is actually different from that of the nlme package (for linear mixed models). In the package bpnreg, the notation “Group + Time” actually considers the possible interaction Group*Time between both factors. We refer the readers to Cremers et & Klugkist (2018) for more details about the use of the bpnreg package.

Cremers J, Klugkist I. One direction? A tutorial for circular data analysis using R with examples in cognitive psychology. Front Psychol. 2018;9:2040.

Pg. 27, line 512: “however, no significant difference…” – I don’t think you need a “however” here. You could just say “there was no significant difference…”.

The word “however” has been removed.

Pg. 33, line 645 – after explaining all the results across the various measures, it might be nice in this first paragraph to more clearly link the theoretical side with the tasks measured. E.g., 1- in the execution of movements (as measured in xx task/s).

This paragraph, L622-628, has been modified as follows:

“The study investigated the rhythmic tapping behavior of people who stutter compared to people who do not stutter and considered several levels of processing at which differences were hypothesized to occur: 1- the execution of movements, in particular their initiation (as measured in the task REACT), 2- the perception of beat, at a given periodicity (as measured in the task ISO_REPRO), 3- the on-line adaptation and improvement of their accuracy and consistency, based on sensory feedback (as measured in the tasks 1:1_ISO_SYNC, 1:4_ISO_SYNC and NONISO_SYNC).”

Pg. 36, Lines 739-743: this sentence is very long and refers to already presented information. This could be written more concisely to avoid having to refer to information “already stated above”.

The sentence on L712-713 has been modified as follows:

“Several arguments were provided in the preceding section (4.1) that exclude the idea that timing differences between PWS and PNS simply result from an impaired motor execution.”

Pg. 40, line 812, do you mean central nervous system?

Thank you for pointing this out. It has been corrected accordingly.

Pg. 41, line 835: do you mean *decreased* PLV in PWS?

Yes., thank you for noticing the mistake. This has been corrected.

---

## [Decision Letter · Decision Letter 2]

12 Oct 2022

Rhythmic tapping difficulties in adults who stutter: a deficit in beat perception, motor execution, or sensorimotor integration?

PONE-D-21-37460R2

Dear Dr. HUEBER,

We’re pleased to inform you that your manuscript has been judged scientifically suitable for publication and will be formally accepted for publication once it meets all outstanding technical requirements.

Kind regards,

Jessica Adrienne Grahn

Academic Editor

PLOS ONE

Additional Editor Comments (optional):

Reviewers' comments:

Reviewer's Responses to Questions

**Comments to the Author**

1. If the authors have adequately addressed your comments raised in a previous round of review and you feel that this manuscript is now acceptable for publication, you may indicate that here to bypass the “Comments to the Author” section, enter your conflict of interest statement in the “Confidential to Editor” section, and submit your "Accept" recommendation.

Reviewer #2: All comments have been addressed

2. Is the manuscript technically sound, and do the data support the conclusions?

Reviewer #2: Yes

3. Has the statistical analysis been performed appropriately and rigorously? 

Reviewer #2: Yes

4. Have the authors made all data underlying the findings in their manuscript fully available?

Reviewer #2: No

5. Is the manuscript presented in an intelligible fashion and written in standard English?

Reviewer #2: Yes

6. Review Comments to the Author

Reviewer #2: All my concerns were addressed. Goodluck!

7. PLOS authors have the option to publish the peer review history of their article (what does this mean?). If published, this will include your full peer review and any attached files.

Reviewer #2: No

---

## [Editor Report · Acceptance letter]

18 Oct 2022

PONE-D-21-37460R2 

Rhythmic tapping difficulties in adults who stutter: a deficit in beat perception, motor execution, or sensorimotor integration? 

Dear Dr. Garnier:

I'm pleased to inform you that your manuscript has been deemed suitable for publication in PLOS ONE. Congratulations! Your manuscript is now with our production department. 

Kind regards, 

on behalf of

Dr Jessica Adrienne Grahn 

Academic Editor

PLOS ONE